# Afterburner: Reinforcement Learning Facilitates Self-Improving Code Efficiency Optimization

**Mingzhe Du**[1,2]    **Luu Anh Tuan**[1]    **Yue Liu**[2]    **Yuhao Qing**[3]    **Dong Huang**[2,3†]

**Xinyi He**[4]    **Qian Liu**[5†]    **Zejun Ma**[5]    **See-kiong Ng**[2]

[1]Nanyang Technological University    [2]National University of Singapore
[3]The University of Hong Kong    [4]Xi'an Jiaotong University    [5]TikTok

{mingzhe001, anhtuan}@ntu.edu.sg, {yliu, dhuang, seekiong}@nus.edu.sg,
yhqing@cs.hku.hk, hxyhxy@stu.xjtu.edu.cn, {qian.liu, mazejun}@tiktok.com

## Abstract

Large Language Models (LLMs) generate functionally correct solutions but often fall short in code efficiency, a critical bottleneck for real-world deployment. In this paper, we introduce a novel test-time iterative optimization framework to address this, employing a closed-loop system where LLMs iteratively refine code based on empirical performance feedback from an execution sandbox. We explore three training strategies: Supervised Fine-Tuning (SFT), Direct Preference Optimization (DPO), and Group Relative Policy Optimization (GRPO). Experiments on our Venus dataset and the APPS benchmark show that SFT and DPO rapidly saturate in efficiency gains. In contrast, GRPO, using reinforcement learning (RL) with execution feedback, continuously optimizes code performance, significantly boosting both PASS@1 (from 47% to 62%) and the likelihood of outperforming human submissions in efficiency (from 31% to 45%). Our work demonstrates effective test-time code efficiency improvement and critically reveals the power of RL in teaching LLMs to truly self-improve code efficiency. We released our code and data at `https://github.com/Elfsong/Afterburner`.

## 1   Introduction

Large Language Models (LLMs) and agent frameworks are catalyzing a profound transformation in software engineering [74, 47, 61, 32, 35, 24, 76], significantly improving the *functional correctness* of their code generation and starting to rival human engineers in certain tasks [7, 69, 28]. However, this focus on correctness often overshadows another critical dimension of software quality: *computational efficiency*. In real-world systems, where latency and memory budgets are paramount, code that is merely *correct but inefficient* can precipitate severe performance bottlenecks, leading to inflated computing costs and system-wide latencies. This chasm between functional correctness and computational efficiency represents a formidable challenge to deploying automatic code generation in mission-critical tasks. This challenge has also spurred the development of code efficiency benchmarks. For instance, EffiBench [26] introduces a relative performance metric against reference solutions, while PIE4PERF [60] utilizes system simulation to meticulously assess the impact of optimizations across a vast corpus of C++ code pairs. Moving beyond pairwise comparisons, Mercury [16] employs percentile ranking against human solutions to highlight the efficiency disparity, and EVALPERF [44] categorizes generated solution efficiency against reference solutions. These benchmarks consistently point out that despite their prowess in generating correct code, current LLMs often produce solutions

---

[†]Corresponding Authors.

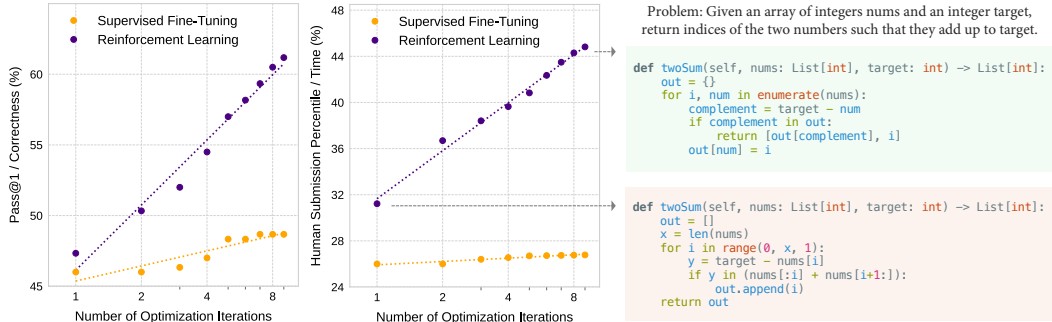

Figure 1: Comparison of iterative optimization performance between a `SFT` model and a `RL` model on `Venus`. While the correctness and efficiency gains of the `SFT` model plateau, the `RL` model facilitates iterative optimization during the inference time effectively.

with suboptimal efficiency [59]. Initial attempts to address this gap, such as Chain-of-Thought [66] in PIE [60], self-optimization in Effilearner [25], or fine-tuning LLMs on an efficiency-oriented dataset [27], have yielded limited success, often failing to instill the adaptive knowledge for robust efficiency improvements.

In this work, we introduce a novel iterative optimization framework (`IOF`) designed to enhance LLM-generated code efficiency through a closed-loop system of *generation and evaluation*, driven by `Afterburner` and `Monolith`. As shown in Figure 2, `Afterburner` takes the original code as input and generates an improved one for the subsequent optimization, where `Monolith` evaluates the improved code and feeds the empirical code performance back to `Afterburner`. The process mirrors how human developers often optimize code through *trial and feedback*.

Our extensive experiments on the novel `Venus` benchmark and the widely-used `APPS` [22] benchmark demonstrate the varied learning dynamics of different optimization strategies within `IOF`. While *Supervised Fine-Tuning (SFT)* [38] offers initial efficiency gains in the first few iterations, it quickly saturates and struggles with sustained improvement. *Direct Preference Optimization (DPO)* [56] consistently performs better than SFT but has the same trend as SFT. In stark contrast, *Group Relative Policy Optimization (GRPO)* [57] continuously refines code performance. As illustrated in Figure 1, it boosts PASS@1 from 47% to 62% and significantly elevates all efficiency metrics, for instance, increasing BEYOND-I from 31% to 45%. We attribute these divergent behaviors to the fundamental nature of what each method tends to capture: **SFT tends to capture superficial patterns** from mimicking examples. **DPO internalizes static preferences** based on pairwise comparisons from offline data. In contrast, through online interaction with execution feedback, **GRPO cultivates an adaptive proficiency** in code efficiency optimization, which enables it to explore and exploit the solution space effectively within an iterative, test-time optimization process. Our key contribution not only lies in demonstrating effective test-time improvement of code efficiency but, more critically, in dissecting how different strategies contribute to this iterative optimization and highlighting the superior adaptability of online feedback-driven RL approaches in efficient-oriented code generation.

## 2 Related Work

**LLMs for Code Generation**   LLMs have demonstrated remarkable progress in code generation, fueled by extensive training on vast code corpora [2, 41, 51, 47]. Building upon foundational models such as Llama [62] and Qwen [70], subsequent efforts have specialized these models for coding tasks, yielding variants like StarCoder [47], QwenCoder [34] and OpenCoder [33]. These models excel in diverse applications, including code completion [8, 38, 12], program repair [50, 45, 73, 46], and unit test generation [29, 3]. Despite their success in generating *functionally correct* code, as evidenced by benchmarks like HumanEval [8], LiveCodeBench [36], and BigCodeBench [77], the *computational efficiency* of the generated code remains a less explored frontier.

**Code Efficiency Evaluation**   Addressing this gap, recent work has focused on quantitatively assessing the efficiency of LLM-generated code [71, 38, 54, 23, 10]. EffiBench [26] collects 1000 efficiency-critical Python problems, evaluating code via an efficiency ratio against reference solutions. PIE4Effi [60] emphasizes the importance of reliable measurement. It utilizes a system simulator for

code execution and contributes a dataset of over 77,000 C++ efficiency preference pairs. Deviating from pairwise comparisons, EVALPERF [44] introduces Differential Performance Evaluation (DPE) on 121 performance-challenging tasks, categorizing generated solution efficiency against reference implementations. Mercury [16] measures efficiency by percentile rank against a substantial corpus of human-written solutions. More recently, ENAMEL [55] proposed an unbiased estimator *eff@k* for time efficiency. These benchmarks reveal that current LLMs still significantly struggle to produce code that consistently matches expert-level computational efficiency. Building on these efforts and inspired by Mercury [16], our work introduces the `Venus` dataset, which expands upon existing resources with more tasks and solutions to facilitate a more rigorous efficiency assessment.

**Preference Alignment in Code Generation**   While *functional correctness* is paramount, *code efficiency* is a critical yet often overlooked preference in LLM-based code generation. Initial attempts to steer LLMs towards efficiency via prompt engineering, such as Chain-of-Thought [66] in PIE [60] or self-optimization in Effilearner [25]. Subsequent instruction tuning methods have predominantly aimed at enhancing functional correctness [48, 66, 67]. Although some recent works like Swift-Coder [27] and PIE4PERF [60] used efficiency-focused datasets for model fine-tuning, their reliance on cross-entropy loss hindered the direct instillation of nuanced efficiency preferences. To achieve finer-grained preference alignment, RL has emerged as a powerful paradigm for code preference alignment [65]. Initial methods like CodeRL [40] use code execution outcomes as feedback. More recent approaches such as StepCoder [14], RLEF [18], and Focused-DPO [72] have significantly advanced *functional correctness* by leveraging execution feedback. However, these RL methods have largely neglected computational efficiency as a primary optimization target, with existing execution environments typically providing only correctness-based rewards. To enable RL-based optimization for *code efficiency*, our work introduces `Monolith`, a high-fidelity sandbox that delivers real-time efficiency metrics, thereby fostering a deeper preference for performant code.

# 3   Iterative Optimization Framework

While current LLMs can produce viable solutions, these often fall short of the performance standards required in resource-constrained or time-sensitive applications [16, 55]. To bridge this gap, we introduce the Iterative Optimization Framework (`IOF`), a novel approach designed to enhance the efficiency of LLM-generated code. As illustrated in Figure 2, `IOF` employs a closed-loop system where code is progressively refined through cycles of *forward generation* and *backward evaluation*.

Central to `IOF` are two synergistic components: **Afterburner**, a model suite that proposes targeted efficiency improvements, and **Monolith**, a robust code execution sandbox that provides precise, real-world performance metrics. The interplay between these components drives each optimization iteration: commencing with an *original code* and an *efficiency instruction*, `Afterburner` takes the inputs to generate an *improved code* alongside its *reasoning content*. This *improved code* is subsequently executed within `Monolith`, yielding empirical efficiency feedback to guide the subsequent optimization iteration. The sections detail the mechanics of `Afterburner` and `Monolith`, and the overall iterative workflow as formalized in Algorithm 1.

## 3.1   Afterburner: Code Efficiency Optimization Models

In the realm of aviation, an afterburner is a secondary combustion system integrated into jet engines, designed to provide a significant thrust augmentation [78]. While this surge in power comes at the cost of considerably higher fuel consumption, it serves as a critical mechanism for scenarios demanding peak performance. Drawing a parallel to this concept, our `Afterburner` aims to push the efficiency of LLM-generated code to the maximum. Instead of consuming more fuel, `Afterburner` leverages the inference-time scaling law[68] and the execution feedback from the `Monolith` sandbox to iteratively refine generated code. For the $i$-th iteration, the process can be formalized as:

$$\mathcal{C}_i^{out} = \texttt{Afterburner}(\mathcal{P}, \mathcal{I}, \mathcal{C}_i^{in}, \mathcal{M}_i^{in}), \qquad (1)$$

where $\mathcal{P}$ is the problem description, $\mathcal{I} \in \{$ *'time-efficient', 'memory-efficient', 'integral-efficient'* $\}$ denotes a specific efficiency instruction (e.g., *minimizing execution time, reducing peak memory usage, or optimizing the integral score*). $\mathcal{C}_i^{in}$ denotes the input solution for the current iteration, and $\mathcal{M}_i^{in} = \texttt{Monolith}(\mathcal{C}_i^{in})$ is its performance metric corresponding to objective $\mathcal{I}$. The refined candidate code $\mathcal{C}_i^{out}$ is then evaluated to obtain its performance metric, $\mathcal{M}_i^{out} = \texttt{Monolith}(\mathcal{C}_i^{out})$.

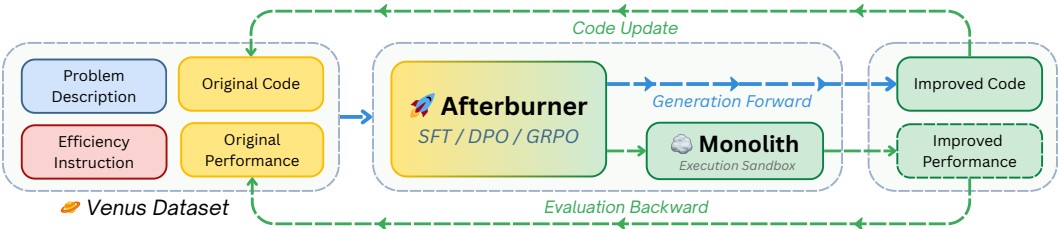

Figure 2: Inference Workflow of the Iterative Optimization Framework (`IOF`). In the forward generation (blue lines), `Afterburner` takes a *problem description*, *efficiency instruction*, *original code (optional)*, and *original performance* as input. It then produces *reasoning content* and *improved code* in a designated format. For the backward evaluation (green lines), the *original code* and *original performance* are updated with the improved versions. The detailed pipeline is defined in Algorithm 1.

For the subsequent iteration, we select the *best-performing* code via a greedy approach:

$$(\mathcal{C}_{i+1}^{in}, \mathcal{M}_{i+1}^{in}) = \begin{cases} (\mathcal{C}_i^{out}, \mathcal{M}_i^{out}) & \text{if } \mathcal{M}_i^{out} \succ \mathcal{M}_i^{in} \\ (\mathcal{C}_i^{in}, \mathcal{M}_i^{in}) & \text{otherwise} \end{cases}, \tag{2}$$

where $\mathcal{M}_i^{out} \succ \mathcal{M}_i^{in}$ indicates that the performance of $\mathcal{C}_i^{out}$ is superior to that of $\mathcal{C}_i^{in}$ with respect to the objective $\mathcal{I}$. The iterative process continues for a predetermined number of iterations $N_{iter}$.

## 3.2 Monolith: Code Execution Sandbox

`Monolith` is a catalyst of `IOF`, which executes generated code and provides the empirical performance feedback to the iterative optimization. Since the efficacy of RL and preference optimization methods hinges on the quality and consistency of the feedback signal [57, 18], `Monolith` prioritizes the consistent measurement in its design. While theoretical complexity analysis (e.g., Big $O$ notation) offers high-level insights into algorithmic scalability [11], it often fails to capture the nuances of real-world performance. A return signal without discrimination may cause the optimization algorithm to lose the optimization gradient [20]. Moreover, Constant factors, specific implementation details (such as language runtime, library choices, and compiler optimizations), and hardware interactions (CPU architecture, memory hierarchy) significantly influence actual execution time and memory consumption [44]. Therefore, for the `Afterburner` models to learn to generate genuinely efficient code, they require empirical metrics from `Monolith` that reflect these practical realities.

$$\{passed, time, memory, integral\} = \texttt{Monolith}(code, test\_cases), \tag{3}$$

where $code$ is the code, $passed$ is a boolean value indicating whether $code$ is passed all test cases. $time$, $memory$, and $integral$ denote the absolute execution time, peak memory usage, and the integral score of the code, respectively. We will explain how to measure these metrics in Section 5.

## 4 Code Efficiency Optimization

**Data Preparation** Recent initiatives like Mercury [16], EffiBench [26], and EVALPERF [44] have made important strides in evaluating code efficiency (see Table 5), but persistent limitations remain. To address these shortcomings, while also building upon these foundational efforts, we introduce **Venus**, a new dataset designed for rigorous code efficiency assessment: (1) Inspired by Mercury [16] and EVALPERF [44], it computes percentile ranks against a diverse distribution of reference solutions, unlike methods relying on single, potentially biased baselines [26, 60]. (2) `Venus` provides a substantially larger set of solutions, averaging 106.6 per task by expanding upon EffiBench [26] and Mercury [16]. This is a significant increase from the fewer than 20 solutions found in existing Python efficiency benchmarks as listed in Table 5, ensuring more stable and reliable percentile calculations. (3) It offers a holistic assessment by evaluating execution time, memory usage, and their combined impact. As shown in Table 7, `Venus` Python set includes 2,181 training and 300 test tasks. From this data, we derived training subsets for various optimization methods:

- **SFT Dataset.** For Supervised Fine-Tuning, $DS_{SFT}$ is constructed by sampling pairs of *functionally correct* solutions for tasks from $\text{Venus}_{train}$, where the solution exhibiting inferior computational efficiency is designated $\mathcal{C}^-$ and the superior one is $\mathcal{C}^+$. $DS_{SFT}$ comprises 58,833 training instances, with 19,611 instances generated for each of the three targeted efficiency instructions.

- **DPO Dataset.** Each instance in the preference dataset $DS_{DPO}$ consists of a prompt $(\mathcal{P}, \mathcal{I}, \mathcal{C}, \mathcal{M})$ and a pair of responses $(\mathcal{C}^+, \mathcal{C}^-)$, where we randomly sample three solutions from $\text{Venus}_{train}$, assigning the best code as $\mathcal{C}^+$ and worst as $\mathcal{C}^-$, and the mediocre $\mathcal{C}^{baseline}$ as the baseline, according to their efficiency performance $\mathcal{M}$ with respect to the objective $\mathcal{I}$. Averaging approximately 13.3K instances per efficiency instruction type, $DS_{DPO}$ contains 90,864 training instances.

- **Cold Start Dataset.** This dataset is designed to rapidly adapt `Afterburner` models to the expected response format. $DS_{COLD}$ is constructed using tasks from $\text{Venus}_{train}$, for which initial responses were generated by *Gemini 2.5 Pro*. From an initial collection of 3,392 raw responses with the '*<thinking><solution>*' format, we filter and construct $DS_{COLD}$ with 2,071 instances.

- **GRPO Dataset.** Since `Afterburner`$_{GRPO}$ learns from code execution feedback, the $DS_{GRPO}$ training dataset does not require ground-truth responses. Each instance herein is a prompt structured as $(\mathcal{P}, \mathcal{I}, \mathcal{C}, \mathcal{M})$. $DS_{GRPO}$ employs all 984 distinct tasks in $\text{Venus}_{train}$.

**Supervised Fine-Tuning**  SFT is the most intuitive approach to imbue LLMs with an initial understanding of code efficiency. Its core idea is to expose the model to the inefficient code paired with the optimized code, thereby teaching it to learn the **patterns** that transform suboptimal solutions into more performant ones. The `Afterburner`$_{SFT}$ takes a prompt $\mathcal{X} = (\mathcal{P}, \mathcal{I}, \mathcal{C}^-, \mathcal{M}^-)$, and the training objective is to minimize the cross-entropy loss for generating the expected response $\mathcal{C}^+$:

$$\mathcal{L}_{SFT}(\pi_\theta) = -\mathbb{E}_{(\mathcal{P}, \mathcal{I}, \mathcal{C}^+, \mathcal{C}^-, \mathcal{M}^-) \sim DS_{SFT}} \left[ \log \pi_\theta(\mathcal{C}^+ | \mathcal{X}) \right], \tag{4}$$

where $\pi_\theta(\mathcal{C}^+ | \mathcal{X})$ is the likelihood of generating the optimized code $\mathcal{C}^+$ given the prompt $\mathcal{X}$. It impels LLMs to learn the mapping from inefficient code to their more efficient counterparts.

**Direct Preference Optimization**  While SFT provides a strong baseline, DPO offers a more direct way to align LLMs with efficiency **preferences** offline, without the need for explicit sampling from a reference model during the training. DPO directly increases the likelihood of positive responses $\mathcal{C}^+$ and decreases that of negative ones $\mathcal{C}^-$, thereby tuning the model to inherently generate more efficient code according to the specified efficiency objective $\mathcal{I}$. Its key advantage is directly optimizing for the preference objective. The `Afterburner`$_{DPO}$ loss function is formulated as:

$$\mathcal{L}_{DPO}(\pi_\theta; \pi_{ref}) = -\mathbb{E}_{(\mathcal{X}, \mathcal{C}^+, \mathcal{C}^-) \sim DS_{DPO}} \left[ \log \sigma \left( \beta \log \frac{\pi_\theta(\mathcal{C}^+ | \mathcal{X})}{\pi_{ref}(\mathcal{C}^+ | \mathcal{X})} - \beta \log \frac{\pi_\theta(\mathcal{C}^- | \mathcal{X})}{\pi_{ref}(\mathcal{C}^- | \mathcal{X})} \right) \right], \tag{5}$$

where $\pi_\theta$ is the target model, $\pi_{ref}$ is a reference model (we use the above `Afterburner`$_{SFT}$ model as the reference). $\mathcal{X} = (\mathcal{P}, \mathcal{I}, \mathcal{C}^{baseline}, \mathcal{M})$ is the input prompt. $\beta$ is a hyperparameter controlling the deviation from the reference model, and $\sigma$ is the logistic function.

**Group Relative Policy Optimization**  Building upon the principles of preference-based learning, GRPO [57] extends the pairwise offline comparison of DPO to a group-wise online ranking scenario. For a given prompt, GRPO generates multiple roll-outs and learns the relative advantage amongst these roll-outs. Inspired by recent works [20, 18], we explore whether it can enhance the code efficiency. As depicted in Figure 4, we first SFT the base model on $DS_{COLD}$ to align it quickly with the designated response format, thereby providing a well-aligned foundation for `Afterburner`$_{GRPO}$.

**Reward Functions.** We encourage `Afterburner`$_{GRPO}$ to think about how to improve the efficiency before generating correct and efficient code. Therefore, the reward function comprises three parts: *format control reward*, *functional correctness reward*, and *computational efficiency reward*:

- **Format Control Reward.** This reward component encourages the model to structure its output in a predefined format. Specifically, `Afterburner` models are expected to have a thinking phase encapsulated in <thinking>...</thinking> tags, followed by the actual code within <solution>...</solution> tags. Eq. (6) defines the reward as 1 when the model response matches the regex pattern (See Appendix D.5), otherwise, the reward will be -1.

$$R_{Format}(C^{out}) = \begin{cases} 1 & \text{if } C^{out} \text{ matches the pattern} \\ -1 & \text{otherwise} \end{cases} \tag{6}$$

- **Functional Correctness Reward.** Ensuring the generated code is functionally sound is paramount. We define a boolean $\mathcal{A} = \texttt{Monolith}(C, test\_cases)$ to indicate whether the provided code $C$ passes all test cases, where $test\_cases$ is a set of test cases. $R_{correct}$ is defined as:

$$R_{correct}(C^{in}, C^{out}) = \begin{cases} 1.0 & \text{if } \mathcal{A}^{out} = 1 \text{ and } \mathcal{A}^{in} = 0 \text{ (upgrade)} \\ 0.5 & \text{if } \mathcal{A}^{out} = 1 \text{ and } \mathcal{A}^{in} = 1 \text{ (maintained passing status)} \\ -0.5 & \text{if } \mathcal{A}^{out} = 0 \text{ and } \mathcal{A}^{in} = 0 \text{ (maintained failing status)} \\ -1.0 & \text{if } \mathcal{A}^{out} = 0 \text{ and } \mathcal{A}^{in} = 1 \text{ (downgrade)} \end{cases} \quad (7)$$

- **Efficiency Improvement Reward.** Given the efficiency instruction $\mathcal{I}$, this reward measures the relative improvement in the corresponding performance metric $\mathcal{E} \in \{time, memory, integral\}$ of a roll-out code compared to the baseline input code. Here, $\mathcal{E} = \texttt{Monolith}(C, test\_cases)$ and $\mathcal{E}_{upper}$ are the absolute performance value and the upper limitation with respect to $\mathcal{I}$, respectively.

$$\mathcal{R}_{efficiency} = \tanh(\mathcal{E}_{gain}), \quad \mathcal{E}_{gain} = \frac{\mathcal{E}_{clip}^{in} - \mathcal{E}_{clip}^{out}}{\mathcal{E}_{clip}^{in} + \epsilon}, \quad \mathcal{E}_{clip} = \text{clip}(\mathcal{E}, 0, \mathcal{E}_{upper}), \quad (8)$$

- **Final Reward.** We apply an additive reward to combine all rewards comprehensively. $\beta_f$, $\beta_e$, and $\beta_c$ are weight hyperparameters to each corresponding reward competent.

$$\mathcal{R}_{final} = \beta_f \cdot \mathcal{R}_{format} + \beta_c \cdot \mathcal{R}_{correct} + \beta_e \cdot \mathcal{R}_{efficiency} \quad (9)$$

**Objective.** GRPO leverages a policy gradient approach to optimize the target policy $\pi_\theta$ based on the old one $\pi_{\theta_{old}}$. The training objective encourages the policy to favor generated candidates that not only possess high intrinsic quality but also demonstrate superior performance relative to their peers within the same generation group for a given prompt. This objective is formalized as:

$$\mathcal{L}_{GRPO}(\pi_\theta; \pi_{\theta_{old}}) = -\mathbb{E}_{\mathcal{X} \sim DS_{GRPO}, \{\mathcal{O}_i\}_{i=1}^G \sim \pi_{\theta_{old}}(\mathcal{O}_i|\mathcal{X})} \left[ \min(\mathcal{W}_i, \text{clip}(\mathcal{W}_i, 1+\epsilon, 1-\epsilon) \cdot \mathcal{A}_i) \right], \quad (10)$$

$$\mathcal{X} = (\mathcal{P}, \mathcal{I}, \mathcal{C}), \quad \mathcal{W}_i = \frac{\pi_\theta(\mathcal{O}_i|\mathcal{X})}{\pi_{\theta_{old}}(\mathcal{O}_i|\mathcal{X})}, \quad \mathcal{A}_i = \frac{\mathcal{R}_i - \text{mean}(\{\mathcal{R}_i\}_{i=1}^G)}{\text{std}(\{\mathcal{R}_i\}_{i=1}^G)}, \quad (11)$$

where $\mathcal{X}$ is the input prompt, $\{O_i\}_{i=1}^G$ is the roll-out group with the size $G$. $\mathcal{W}_i$ denotes the policy ratio comparing how the new policy $\pi_\theta$ prefer a generation against the old policy $\pi_{\theta_{old}}$. To prevent drastic $\mathcal{W}_i$ updates, we clip the ratio within the interval $[1 - \epsilon, 1 + \epsilon]$. Finally, $\mathcal{A}_i$ is computed on the reward score of each roll-out $\mathcal{R}_i$, to show the relative advantage in the same roll-out group.

## 5 Experiment Setup

**Dataset Recipe** $\texttt{Venus}$ Python subset contains *2,181* algorithmic problems, each accompanied by a validated test case generator and an average of *106.6* human solutions, enabling robust empirical analysis of *code efficiency* beyond *functional correctness*. For each LLM-generated test case input, we follow the paradigm of Mercury [16], where we execute them through all collected solutions from LeetCode, and only keep those cases having consistent outputs over all correct solutions [30]. Based on $\texttt{Venus}$, Section 4 introduces several datasets for $\texttt{Afterburner}$ training, including $DS_{SFT}$, $DS_{DPO}$, $DS_{COLD}$, and $DS_{GRPO}$. $\texttt{APPS}$ is a widely recognized benchmark for evaluating the functional correctness of code generation models [22]. While its original design, with 21.2 test cases and 23.4 solutions per problem, focuses on correctness, we integrate it into our efficiency evaluation pipeline as an auxiliary benchmark (see Appendix B).

**Functional Correctness** Ensuring functional correctness is a prerequisite for code generation models. Following the evaluation paradigm in Codex [9], we employ the PASS@1 $= N_{passed}/N_{total}$ score to assess the global functional correctness, where $N_{passed}$ is the number of passed generations and $N_{total}$ is the total number of test tasks.

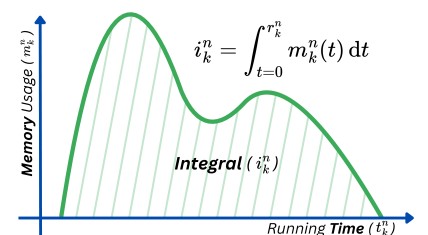

Figure 3: Illustration of task-level efficiency metrics.

$$\text{PR}(x, D) = \frac{1}{|D|} \sum_{d \in D} \mathbf{1}[d \geq x]. \quad (12)$$

$$\text{BEYOND-}\{\text{T, M, I}\} = \frac{\sum_{k=1}^{|V|} \text{PR}(\mathcal{E}_k^{gen}, \{D_k^T, D_k^M, D_k^I\})}{|V_{test}|}, \quad (13)$$

**Computational Efficiency** Following Mercury [16] and EffiBench [26], we avoid employing *absolute* efficiency metrics because they are highly sensitive to hardware configurations and operating systems. For each task in Venus test set $v_k \in V_{test}$, we instead compute *percentile ranks* of an absolute performance $\mathcal{E}_k^{gen}$ relative to the distribution $D_k$ collected from corresponding reference solutions $S_k$. Except the *execution time* ($r_k^{gen}$) and *peak memory* ($\max(m_k^{gen}(t))$), we also consider using the *integral* score $i_k^{gen} = \int_{t=0}^{r_k^{gen}} m_k^{gen}(t)\,\mathrm{d}t$ as a comprehensive efficiency metric, where $m_k^{gen}(t)$ is the instantaneous memory footprint at time $t$. To compute relative efficiency metrics, we establish reference distributions of execution time overhead $D_k^T = \{r_k^n\}_{n=1}^{|S_k|}$, memory overhead $D_k^M = \{m_k^n\}_{n=1}^{|S_k|}$, and integral efficiency $D_k^I = \{i_k^n\}_{n=1}^{|S_k|}$, where $r_k^n$, $m_k^n$, and $i_k^n$ are the absolute execution time, memory usage, and integral score of the $n$-th collected solution $s_k^n \in S_k$, respectively. Based on these distributions, we can calculate the task-level efficiency percentile-rank of the generated solution in Eq. (12). The global efficiency metrics are computed as the average of all task-level percentile-ranks in Eq. (13). **Higher scores indicate that the generated code outperforms a larger fraction of the reference solutions, reflecting stronger code efficiency.**

**Implementation Details** Afterburner models are trained on a single node with eight H100 GPUs. We utilized Llama-Factory [75] for SFT and DPO training phases, and Verl [58] for GRPO training. Dataset construction details can be found in Section 5. For inference acceleration, we use vLLM [39]. Comprehensive details regarding the training pipeline (as shown in Figure 4) and hyperparameters are provided in the Appendix D. Monolith configuration can be found in Appendx H.

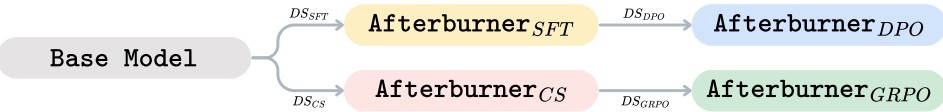

Figure 4: Illustration of the training pipeline of Afterburner models.

## 6 Discussion and Key Takeaways

### 6.1 How about the Code Efficiency Performance of Vanilla LLMs?

Our baseline evaluation of diverse LLMs on the Venus and APPS benchmarks (Tables 1 and 9) reveals a critical performance limitation: **Despite achieving high functional correctness (PASS@1), vanilla models generate code with strikingly inferior computational efficiency compared to human solutions** [44, 55]. For example, *OpenAI o4 mini*, a top-performing model with 89.11% PASS@1 on Venus, produces code whose runtime efficiency (BEYOND-T) surpasses only 56.85% of human solutions (and merely 40.07% on APPS), with similar disparities observed for other leading models and across all efficiency metrics. While stronger (bigger) models exhibit marginally better code efficiency, this is insufficient to overcome the fundamental gap. This pervasive efficiency deficit in LLM-generated code clearly motivates the development of dedicated optimization frameworks, such as Afterburner, to enhance code generation in real-world applications.

### 6.2 Does Iterative Improvement Framework Work?

The foundational hypothesis of the Afterburner framework is that iterative refinement, driven by execution feedback, can progressively enhance code efficiency. This section investigates the effectiveness of such iterative self-optimization and how the choice of underlying optimization strategy impacts learning dynamics and outcomes across successive iterations. Notably, the prompt placeholder *original_code* is left empty for the initial code generation (see Section E).

- **SFT Memorized Superficial Patterns.** SFT primarily learns to mimic transformations from less to more efficient code based on its training data. In the model training phase, Afterburner$_{SFT}$ updates these learned *patterns*. Initial gains are possible if the input code matches known suboptimal patterns. However, SFT's capacity to generalize to novel inefficiencies or explore fundamentally different algorithmic solutions is inherently limited, as it lacks a deep understanding of *why* a pattern is efficient beyond its training data co-occurrence. Consequently, as seen in Figure 5, SFT often quickly exhausts its applicable patterns in iterative optimization.

Table 1: Comparison of Vanilla Efficiency Performance between Open-Source and Closed-Source Models on the Venus Benchmark. Parentheses denote 95% CI. The top score for each metric is highlighted in **bold**. `Afterburner` uses *'both time and memory efficient'* instruction in the generation.

| Model Name | PASS@1 ↑ | BEYOND-T ↑ | BEYOND-M ↑ | BEYOND-I ↑ |
|---|---|---|---|---|
| *Open-source Models* | | | | |
| Qwen 2.5 3B | 27.99 | 12.40 (12.35, 12.45) | 13.24 (13.21, 13.28) | 10.29 (10.24, 10.34) |
| Qwen 2.5 Coder 7B | 52.21 | 20.66 (20.61, 20.71) | 25.21 (25.16, 25.26) | 16.78 (16.74, 16.83) |
| Qwen 2.5 7B instruct | 60.78 | 27.67 (27.61, 27.73) | 29.79 (29.73, 29.85) | 21.02 (20.98, 21.07) |
| Llama 4 Scout | 62.82 | 33.10 (33.03, 33.16) | 38.22 (38.17, 38.26) | 26.91 (26.86, 26.95) |
| DeepSeek V3 | 86.33 | 48.66 (48.57, 48.75) | 51.20 (51.15, 51.25) | 39.20 (39.13, 39.26) |
| QwQ 32B ⋄ | 83.09 | 51.09 (51.03, 51.16) | 45.22 (45.16, 45.27) | 41.66 (41.61, 41.70) |
| *Closed-source Models* | | | | |
| OpenAI 4o | 82.26 | 38.22 (38.15, 38.29) | 42.09 (42.04, 42.15) | 28.89 (28.84, 28.95) |
| Claude 3.5 Haiku | 66.45 | 38.82 (38.75, 38.89) | 37.77 (37.71, 37.82) | 30.15 (30.10, 30.20) |
| Claude 3.7 Sonnet | 86.52 | 52.19 (52.10, 52.27) | 49.86 (49.81, 49.92) | 40.49 (40.43, 40.55) |
| OpenAI o4 mini ⋄ | **89.11** | **56.85** (56.77, 56.93) | **53.41** (53.35, 53.46) | **45.71** (45.66, 45.77) |
| *Our Afterburner Tuned on Qwen 2.5 3B at Iteration 10* | | | | |
| `Afterburner`$_{SFT}$ | 48.67 | 26.78 (26.72, 26.91) | 25.30 (25.25, 25.41) | 22.50 (22.41, 22.67) |
| `Afterburner`$_{GRPO}$ | 61.67 | 45.17 (45.08, 45.30) | 48.05 (47.96, 48.26) | 38.95 (38.89, 39.17) |

- **DPO Realized Static Preferences.** DPO internalizes *preferences* for more efficient solutions from ranked pairs. This allows `Afterburner`$_{DPO}$ to make more nuanced judgments than SFT, guided by characteristics correlated with better performance under the objective $\mathcal{I}$. Iteratively, DPO can steer code towards these preferred traits. However, since DPO is typically an offline method, it does not learn from its own generations without retraining. Thus, its exploration is still bounded by the diversity of its initial preference dataset. Figure 5 shows DPO may offer more consistent improvement than SFT, but also tends to plateau once its learned preferences are fully exploited.

- **GRPO Cultivated Adaptive Proficiency.** GRPO utilizes an online reinforcement learning approach. In the training phase, `Afterburner`$_{GRPO}$ generates multiple candidates, which are evaluated by `Monolith`. The resultant empirical feedback directly updates the policy $\pi_\theta$ to favor strategies yielding more efficient code for objective $\mathcal{I}$. This online learning is pivotal for iterative self-improving optimization. Rather than merely static *patterns* or *preferences*, GRPO develops a deeper *proficiency* in code optimization. By actively exploring the solution space and receiving direct feedback, `Afterburner`$_{GRPO}$ continuously refines its generation strategy, adapts to problem-specific nuances, and uncovers sophisticated optimization policy over iterations. The group-wise ranking further enhances its fine-grained understanding of relative efficiencies. This adaptive capability, evident in Figure 5, allows GRPO to achieve sustained and superior performance improvements, continually pushing its optimization boundaries.

- **Model Generalization.** To verify whether our models can generalize to out-of-distribution questions, we evaluated Afterburner on the APPS [22]. As illustrated in Appendix Figure 9, `Afterburner` demonstrates a similar pattern of performance improvement, confirming its effectiveness on problems with distinct data distribution.

## 6.3 Why GRPO Can Iteratively Enhance Code Efficiency?

**Generation diversity is foundational to its iterative capability**. By unleashing the KL divergence restriction in the training phase, `Afterburner`$_{GRPO}$ inherently explores multiple potential optimization pathways without the ground-truth. This diversity ensures that `Afterburner`$_{GRPO}$ is not confined to local optima. Moreover, GRPO **gains experience improving code from what it generated through the iterative refinement loop**. It does not just generate code, but executes it to gather concrete feedback on its real-world performance, effectively learning from its successes and failures in a continuous cycle. As the model identifies more efficient code structures in training, it becomes progressively better at producing them in inference. Ablation studies (Table 2) confirm that removing the feedback mechanism or original code context significantly diminishes `Afterburner`$_{GRPO}$ performance, an effect not always as evident in `Afterburner`$_{SFT}$ or `Afterburner`$_{DPO}$.

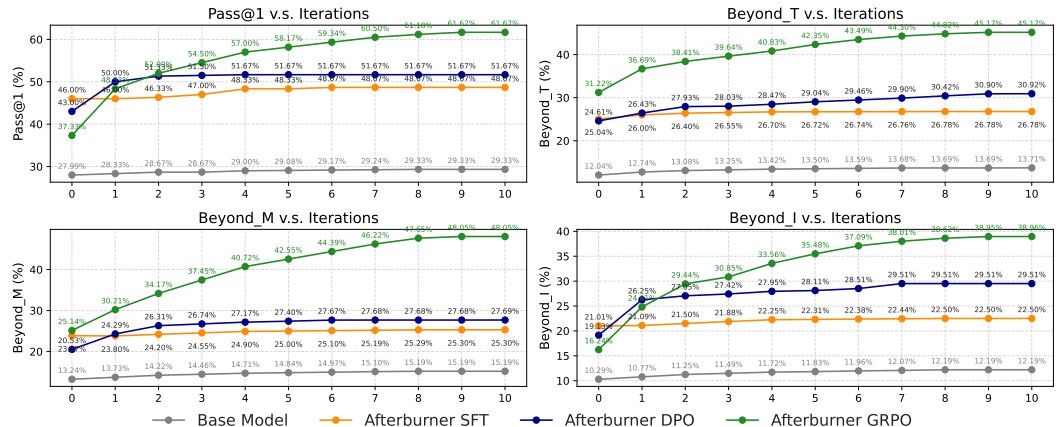

Figure 5: Iterative Optimization with an Efficient Instruction *'both time and memory efficient'*.

Table 2: Performance of `Afterburner` models at Iteration 4 with removing execution feedback and original code input, respectively. Bracketed values represent the change in performance compared to the baseline: red indicates degradation, and green indicates improvement.

| Model/Method | PASS@1 | BEYOND-T | BEYOND-M | BEYOND-I |
|---|---|---|---|---|
| **Afterburner-SFT** | 48.33 | 26.61 | 24.39 | 22.25 |
| - Remove Feedback | 46.33 (-2.00) | 25.41 (-1.20) | 24.70 (+0.31) | 21.43 (-0.82) |
| - Remove Original Code | 45.33 (-3.00) | 25.64 (-0.97) | 26.17 (+1.78) | 20.08 (-2.17) |
| **Afterburner-DPO** | 51.67 | 28.45 | 28.03 | 27.89 |
| - Remove Feedback | 50.33 (-1.34) | 27.33 (-1.12) | 26.73 (-1.30) | 25.68 (-2.21) |
| - Remove Original Code | 47.33 (-4.34) | 25.32 (-3.13) | 24.17 (-3.86) | 22.01 (-5.88) |
| **Afterburner-GRPO** | 57.00 | 40.81 | 40.68 | 33.51 |
| - Remove Feedback | 52.51 (-4.49) | 34.15 (-6.66) | 34.49 (-6.19) | 29.87 (-3.64) |
| - Remove Original Code | 54.17 (-2.83) | 32.17 (-8.64) | 33.25 (-7.43) | 24.24 (-9.27) |

### 6.4 Can Afterburner Generate Code Surpassing Human Efficiency?

While LLMs excel at generating functionally correct code, often by imitating human-written examples in their training data, a key question remains: **Can they produce solutions exceeding the code efficiency of this best human-written code?** To investigate this, we compare the efficiency of model-generated code against human solutions from `Venus`. As presented in Table 3, reasoning models such as *QwQ 32B* and *OpenAI o4-mini* exhibit a higher ability to occasionally generate super-human solutions. Crucially, our proposed $\text{Afterburner}_{GRPO}$ yields the highest B% scores across all evaluated metrics after 8 iterations: TIME (8.00%), MEMORY (7.00%), and INTEGRAL (5.33%). This demonstrates that $\text{Afterburner}_{GRPO}$ moves beyond merely replicating common patterns observed during pre-training. By actively exploring the solution space through RL, it discovers highly optimized implementations that are often structurally different from canonical human approaches. However, this enhanced exploration entails a trade-off: $\text{Afterburner}_{GRPO}$ also generates a larger fraction of solutions that are less efficient than the human baseline.

## 7 Conclusion

We introduced an iterative optimization framework designed to enhance the computational efficiency of LLM-generated code. Central to this framework are the `Afterburner` models, which are critically guided by real-time efficiency feedback from the `Monolith` sandbox. Our comparative analysis of distinct optimization strategies revealed that SFT primarily learned superficial code optimization patterns, while DPO internalized efficiency preferences. In stark contrast, by leveraging online RL with direct execution feedback, GRPO achieved superior and sustained improvements in code efficiency across multiple iterations.

Table 3: Model vs. Human on Venus. **Bold** indicates the top performance per column and model category. B%, M%, W%, and F% denote percentages of solutions: **Better** than all human, Within **mediocre** human range, **Worse** than all human, or **Failed** to pass all test cases, respectively.

| Model Name | Time | | | | Memory | | | | Integral | | | |
|---|---|---|---|---|---|---|---|---|---|---|---|---|
| | B% | M% | W% | F% | B% | M% | W% | F% | B% | M% | W% | F% |
| Qwen 2.5 3B | 0.67 | 27.00 | 0.33 | **72.00** | 0.33 | 27.33 | 0.33 | **72.00** | 0.67 | 26.67 | 0.67 | **72.00** |
| Qwen 2.5 Coder 7B | 1.33 | 50.67 | 0.33 | 47.67 | 0.67 | 50.67 | 1.00 | 47.67 | 1.33 | 50.67 | 0.33 | 47.67 |
| Qwen 2.5 7B Instruct | 1.67 | 58.33 | **0.67** | 39.33 | 1.00 | 58.33 | **1.33** | 39.33 | 1.33 | 58.00 | **1.67** | 39.33 |
| Llama 4 Scout Instruct | 3.00 | 59.33 | 0.33 | 37.33 | 2.00 | 60.67 | 0.33 | 37.33 | 1.67 | 60.67 | 0.67 | 37.33 |
| Deepseek V3 | 5.33 | 80.67 | 0.67 | 13.67 | **3.33** | 82.67 | 0.33 | 13.67 | 3.00 | 81.67 | **1.67** | 13.67 |
| QwQ 32B | **6.67** | **76.00** | 0.33 | 17.00 | 2.33 | **79.67** | 1.00 | 17.00 | **3.33** | **79.00** | 1.00 | 17.00 |
| GPT-4o | 2.33 | 79.00 | **1.00** | 17.67 | 1.33 | 79.00 | **1.67** | 17.67 | 1.33 | 79.67 | 1.33 | 17.67 |
| Claude 3.5 Haiku | 4.67 | 61.67 | 0.33 | **33.67** | 2.00 | 64.00 | 0.33 | **33.67** | 2.67 | 63.33 | 0.67 | **33.67** |
| Claude 3.7 Sonnet | 5.67 | 80.67 | 0.33 | 13.33 | 2.67 | 83.33 | 0.33 | 13.33 | 3.33 | 82.00 | 1.00 | 13.33 |
| O4-mini | **7.00** | **82.00** | 0.00 | 11.00 | **3.33** | **85.33** | 0.67 | 11.00 | **4.00** | **84.33** | 0.67 | 11.00 |
| Afterburner$_{GRPO}$ | **8.00** | **46.33** | **7.33** | 38.33 | **7.00** | **44.33** | **10.33** | 38.33 | **5.33** | **46.00** | **10.00** | 38.33 |

# 8 Limitations

While `Afterburner` demonstrates effective efficiency optimization for competition-level programming tasks, its extension to larger, real-world software engineering projects warrants further investigation. These projects often entail greater complexity in their code context, diverse efficiency criteria beyond algorithmic performance (e.g., library interactions or I/O operations), and may require sophisticated strategies for task decomposition, which are outside the scope of the current work.

Moreover, our iterative optimization framework inherently requires more inference time during the code generation phase compared to single-pass methods. We argue that this upfront investment in optimization can be offset by significant cumulative runtime savings when the highly efficient code is deployed in production, especially for frequently executed or performance-critical modules. Nonetheless, this trade-off between the optimization cost and long-term execution benefits needs to be carefully evaluated based on specific application requirements and deployment scenarios.

# 9 Acknowledgment

This research is supported by DSO grant DSOCL23216. This research is also supported by A*STAR, CISCO Systems (USA) Pte. Ltd and National University of Singapore under its Cisco-NUS Accelerated Digital Economy Corporate Laboratory (Award I21001E0002).

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

# A  Model Details

| Model Name | Model Size | URL |
|---|---|---|
| Qwen 2.5 3B [70] | 3B | https://huggingface.co/Qwen/Qwen2.5-3B |
| Qwen 2.5 Coder 7B [70] | 7B | https://huggingface.co/Qwen/Qwen2.5-Coder-7B-Instruct |
| Qwen 2.5 7B [70] | 7B | https://huggingface.co/Qwen/Qwen2.5-7B-Instruct |
| Llama 4 Scout 17B 16E Instruct [49] | 17B | https://huggingface.co/meta-llama/Llama-4-Scout-17B-16E-Instruct |
| QwQ 32B [70] ◇ | 32B | https://huggingface.co/Qwen/QwQ-32B |
| GPT-4o [1] | Unknown | https://platform.openai.com/docs/models/gpt-4o |
| Claude 3.5 Haiku [4] | Unknown | https://www.anthropic.com/claude/haiku |
| Claude 3.7 Sonnet [5] | Unknown | https://www.anthropic.com/claude/sonnet |
| DeepSeek V3 [42] | Unknown | https://www.deepseek.com/ |
| O4-mini [52] ◇ | Unknown | https://platform.openai.com/docs/models/o4-mini |

Table 4: Model list with their size, reasoning ability, and model URL. ◇ denotes a reasoning model.

# B  Dataset Curation and Statistics

Table 5: Overall Statistics of Representative Function-level Code Generation Benchmark Datasets. ♣ indicates the datasets are designed for functional correctness solely, while ♡ indicates the datasets are designed for code efficiency. We list the average number of solutions per problem for each dataset. The detailed definition of each metric can be found in Section 2. ∗ indicates that there are some works [74] that extend the original datasets to more diverse programming languages.

| Dataset | Tasks | Test Cases | Solutions | Metrics | Languages | Source |
|---|---|---|---|---|---|---|
| ♣ HumanEval [8] | 164 | 8.1 | 1.0 | Pass@k | ∗ Python | Crowdsource |
| ♣ MBPP [6] | 257 | 3.0 | 1.0 | Pass@k | Python | Crowdsource |
| ♣ APPS [22] | 10,000 | 21.2 | 23.4 | Pass@k | ∗ Python | CodeForces |
| ♣ BigCodeBench [77] | 1,140 | 5.6 | 1.0 | Pass@k | Python | Synthesis |
| ♡ EffiBench [26] | 1000 | 100 | 14.6 | NET/ NMU | Python | LeetCode |
| ♡ Mercury [16] | 1,889 | $+\infty$ | 18.4 | Pass/ Beyond | Python | LeetCode |
| ♡ ENAMEL [55] | 142 | 20 | 1 | Eff@k | Python | HumanEval |
| ♡ EVALPERF [44] | 1,474 | − | 10 | DPS | Python | [8, 6, 22, 43] |
| ♡ PIE [60] | 1,889 | 104 | 80.6 | %Opt / %Correct / Speedup | CPP | CodeNet |
| ♡ ECCO [64] | 48 | 20 | 16.5 | Time/Memory | python | CodeNet |
| ♡ Venus (ours) | 8,598 | $+\infty$ | 79.3 | Pass/ Time/ Memory/ Integral | Multilingual | LeetCode |

## B.1  Venus Dataset

We constructed the Venus benchmark through a multi-stage filtering pipeline, as illustrated in Figure 6. Beginning with 3,535 problems from LeetCode [†], we first removed paid-only questions to adhere to fair-use principles, retaining 2,821 freely accessible problems. We then filtered for *algorithmic* problems, discarding other categories such as *Database* or *Shell*. To ensure reliable efficiency distribution in the evaluation, we executed all available solutions using the Monolith runtime. Problems with fewer than 16 solutions passing all test cases were further excluded. This resulted in a curated set of 1,284 high-quality problems. Finally, we split the dataset into a training set of 984 problems and a held-out test set of 300 problems, forming the complete Venus dataset.

**Multilingual Scope.**  Transcending the prevalent Python focus of prior benchmarks [8, 6, 77, 26, 16, 21], Venus offers robust support for a multilingual scope as listed in Table 7. Since the test case generator is rooted in standard I/O-based test case interaction [31], Venus can further support multilingual code generation benchmarking and training.

**Language-Agnostic Test Cases**  A significant challenge in benchmarking code generation models, especially for efficiency, is the availability of extensive and diverse test cases. Most online judge platforms do not disclose their test suites, and existing benchmarks often provide a limited number of test cases (Table 5), which may be insufficient for robust efficiency profiling. To address this issue, we propose a novel approach to generate a large-scale, language-agnostic test case dataset. The process involves:

---

[†] https://leetcode.com/problemset/

Table 6: Definitions of the fields within `Venus` datasets.

| Column Name | Description |
|---|---|
| problem_id | Unique identifier for each problem (int64) |
| title | Title of the problem (string) |
| question_content | Full text of the problem statement (string) |
| difficulty | Difficulty level (categorical) |
| tags | List of associated tags (sequence) |
| code_prompt | Prompt used for solution generation (string) |
| test_case_generator | Code generating test cases (string) |
| test_case_evaluator | Code evaluating test case outputs (string) |
| test_case_runners | Code executing solutions with test cases (string) |
| solutions | Human-submitted solutions from LeetCode (list of strings) |

Figure 6: Pipeline for constructing the **Venus** dataset. We start from 3,535 LeetCode problems and apply a series of quality-control and de-duplication filters, retaining 1,284 high-quality problems in the Venus benchmark.

- **Automated Test Case Generation:** For each problem in `Venus`, a dedicated test case generator program is synthesized by GPT-4O based on the given problem description. These generators are designed to produce a virtually unbounded stream of diverse and valid inputs.

- **Rigorous Validation:** The validity of each generated test case is paramount. Before being used for evaluation, a candidate test case is run against all collected canonical human solutions for that problem. Only test cases for which all canonical solutions produce consistent outputs are accepted. This ensures that the test cases are unambiguous and accurately reflect the problem's requirements as understood by proficient human programmers.

- **Standard I/O for Language Agnosticism:** The key to the multilingual capability of `Monolith` lies in its interaction protocol with the code being tested. All solutions, irrespective of their programming language, interface with the test harness exclusively via standard input (*stdin*) and standard output (*stdout*). Test inputs are provided as text streams via *stdin*, and the solution's output is captured from *stdout*. This text-based I/O mechanism decouples the test data from the specifics of any programming language.

This design allows the same set of validated test cases to be used for evaluating solutions written in any of the languages supported by `Venus` (Python, C++, Go, Java, JavaScript, etc., as shown in Table 7). This language-agnostic approach not only broadens the applicability of our framework but also simplifies its extension to new programming languages in the future, as new test case generators are not required for each language. The common testbed ensures fair and consistent efficiency comparisons across different languages and models.

**Venus Justification.** **Multilingual Coverage.** A primary contribution is that Venus is the first multilingual code efficiency benchmark, covering six languages. Prior work, including Mercury [16], EffiBench [26], and EvalPerf [44], focuses exclusively on Python. EffiBench-X [53] is a multilingual

Table 7: Breakdown of `Venus` dataset by programming language. For each language we list the total number of tasks and the average number of human submission per task.

| Language | Python | C++ | Go | Java | JavaScript | Total |
|---|---|---|---|---|---|---|
| **Train Tasks** | 2,181 | 2,183 | 866 | 1,358 | 704 | 7,298 |
| **Test Tasks** | 300 | 300 | 200 | 300 | 200 | 1,300 |
| **Avg. Solutions** | 106.6 | 112.2 | 33.6 | 69.6 | 74.4 | 79.3 |

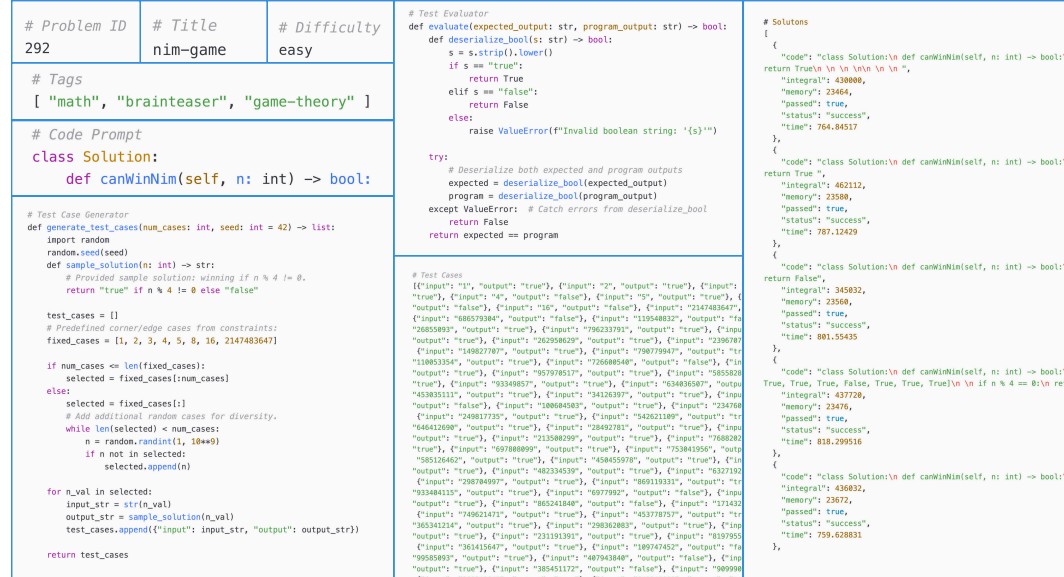

Figure 7: An Example in `Venus` Python Subset.

efficiency benchmark at the same time. **Solutions Diversity.** Venus offers a significantly larger and more diverse set of reference solutions. While EffiBench uses a single baseline and Mercury averages 18.4 solutions per problem, Venus provides an average of 79.3 solutions per problem for each language. This massive increase directly contributes to the statistical diversity of performance metrics, as shown in the time and memory distributions in Appendix Figure 7. **Evaluation Dimensions.** As a direct extension of Mercury, which only measures execution time, Venus evaluates execution time, memory usage, and their integral, providing a more holistic assessment of code efficiency. To better illustrate the **Venus** dataset, we provide a complete instance from **Venus** as shown in Figure 7.

## B.2 APPS Dataset

APPS is a widely recognized benchmark for evaluating the functional correctness of code generation models [22]. While its original design focuses on correctness, we integrate it into our efficiency evaluation pipeline as an auxiliary benchmark. It consists of *10,000* Python programming problems, where each problem is accompanied by an average of *21.2* test cases and *23.4* solutions.

Table 8: Definitions of the fields within `APPS` datasets.

| Column Name | Description |
|---|---|
| `title` | Title of the problem (string) |
| `question_content` | Full text of the problem statement (string) |
| `difficulty` | Difficulty level (categorical) |
| `solutions` | Human-submitted solutions from LeetCode (list of strings) |
| `test_cases` | Test cases (list of strings) |

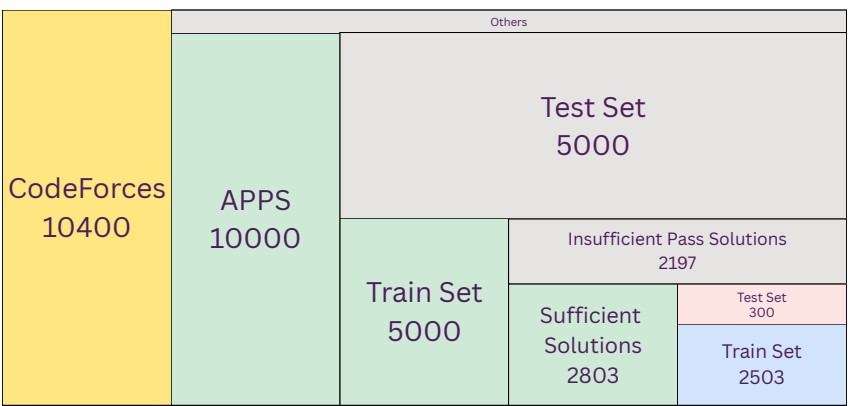

Figure 8: Selection procedure for the **APPS** subset used in our benchmark. Beginning with the official APPS training split (5,000 problems), we discard problems that lack a sufficient number of accepted reference solutions, yielding 2,803 problems in the final dataset.

Table 9: Comparison of Efficiency Performance between Open-Source and Closed-Source Models on the APPS Benchmark. Parentheses denote 95% confidence intervals. The top score for each metric is highlighted in **bold**, while the leading score within each model category is underlined.

| Model Name | Pass@1 | Beyond-T | Beyond-M | Beyond-I |
|---|---|---|---|---|
| *Open-source Models* | | | | |
| Qwen 2.5 3B | 9.67 | 5.10 (5.06, 5.13) | 5.48 (5.45, 5.51) | 3.80 (3.78, 3.83) |
| Qwen 2.5 Coder 7B | 16.00 | 9.30 (9.25, 9.34) | 8.79 (8.75, 8.83) | 6.81 (6.78, 6.85) |
| Qwen 2.5 7B instruct | 17.00 | 9.65 (9.59, 9.70) | 9.33 (9.29, 9.37) | 7.28 (7.24, 7.32) |
| Llama 4 Scout | 47.67 | 24.85 (24.78, 24.92) | 27.96 (27.90, 28.03) | 18.22 (18.17, 18.28) |
| QwQ 32B ◇ | 69.62 | 35.59 (35.51, 35.68) | 37.84 (37.75, 37.93) | 27.32 (27.26, 27.39) |
| *Closed-source Models* | | | | |
| GPT-4o | 46.23 | 26.30 (26.24, 26.37) | 26.47 (26.41, 26.53) | 19.76 (19.71, 19.82) |
| Claude 3.5 Haiku | 36.67 | 13.63 (13.57, 13.69) | 20.96 (20.91, 21.01) | 10.74 (10.70, 10.77) |
| Claude 3.7 Sonnet | 45.63 | 21.68 (21.61, 21.75) | 25.76 (25.70, 25.81) | 15.75 (15.69, 15.81) |
| DeepSeek V3 | 56.63 | 31.58 (31.49, 31.66) | 31.42 (31.35, 31.49) | 23.20 (23.13, 23.27) |
| OpenAI o4 Mini ◇ | **78.81** | **40.07** (39.98, 40.15) | **41.00** (40.93, 41.07) | **28.76** (28.69, 28.82) |

## C Iterative Efficiency Optimization Procedure

The Iterative Efficiency Optimization Procedure, detailed in Algorithm 1, is designed to systematically enhance source code performance. Given a problem description $\mathcal{P}$, an efficiency instruction $\mathcal{I}$ (e.g., targeting time or memory), and a set of test cases $T_{cases}$, the algorithm iteratively refines code. It begins with an initial code version $\mathcal{C}_0^{in}$, which is generated by the `Afterburner` component if not provided, and its initial performance $\mathcal{M}_0^{in}$ is evaluated by the `Monolith` component. Over $N_{iter}$ iterations, new code versions are proposed by `Afterburner` based on the current code and its metrics, and then evaluated by `Monolith`. If a newly generated version $\mathcal{C}_i^{out}$ exhibits improved performance $\mathcal{M}_i^{out}$ according to the criterion $\mathcal{I}$ when compared to the current iteration's input performance $\mathcal{M}_i^{in}$, it is adopted as the input for the subsequent iteration; otherwise, the previous code is retained. The procedure concludes by returning the best-performing code $\mathcal{C}_{N_{iter}}^{in}$ found after $N_{iter}$ iterations, along with its corresponding performance metrics $\mathcal{M}_{N_{iter}}^{in}$.

## D Model Training Details

### D.1 Training Pipeline.

As shown in Figure 4, we explore different optimization strategies to train the `Afterburner` models. Initially, `Afterburner`$_{SFT}$ models are trained using the $DS_{SFT}$ dataset. For `Afterburner`$_{DPO}$ models, we initialize them from the checkpoints of the corresponding `Afterburner`$_{SFT}$ models and

**Algorithm 1** Iterative Efficiency Optimization Procedure

---

**Input**: Problem description $\mathcal{P}$, Efficiency instruction $\mathcal{I} \in \{$*time, memory, integral*$\}$, Set of test cases $T_{cases}$, Original code $\mathcal{C}_0^{in}$ (optional), Number of iterations $N_{iter}$

**Output**: Improved code $\mathcal{C}_0^{out}$, Improved code performance $\mathcal{M}_0^{out}$

---

    **if** not $\mathcal{C}_0^{in}$ **then**

        $\mathcal{C}_0^{in} \leftarrow \texttt{Afterburner}(\mathcal{P}, \mathcal{I}, None, None)$                                $\triangleright$ Initial code generation.

    **end if**

    $\mathcal{M}_0^{in} \leftarrow \texttt{Monolith}(\mathcal{C}_0^{in}, T_{cases})$                                  $\triangleright$ Initial code evaluation.

    **for** $i \leftarrow 1$ **to** $N_{iter}$ **do**

        $\mathcal{C}_i^{out} \leftarrow \texttt{Afterburner}(\mathcal{P}, \mathcal{I}, \mathcal{C}_i^{in}, \mathcal{M}_i^{in})$                      $\triangleright$ Code optimization.

        $\mathcal{M}_i^{out} \leftarrow \texttt{Monolith}(\mathcal{C}_i^{out}, T_{cases})$                          $\triangleright$ Code evaluate.

        **if** $\mathcal{M}_i^{out} \succ \mathcal{M}_i^{in}$) **then**                  $\triangleright$ Compare the performance concerning $I$.

            $(\mathcal{C}_{i+1}^{in}, \mathcal{M}_{i+1}^{in}) \leftarrow (\mathcal{C}_i^{out}, \mathcal{M}_i^{out})$          $\triangleright$ Update with the better performing candidate.

        **else**

            $(\mathcal{C}_{i+1}^{in}, \mathcal{M}_{i+1}^{in}) \leftarrow (\mathcal{C}_i^{in}, \mathcal{M}_i^{in})$               $\triangleright$ Otherwise, retain the current best.

        **end if**

    **end for**

    **return** $(\mathcal{C}_{N_{iter}}^{in}, \mathcal{M}_{N_{iter}}^{in})$      $\triangleright$ Return the best code found after $N_{iter}$ iterations and its metrics

---

subsequently finetune them on the $DS_{DPO}$ dataset. The training process for $\texttt{Afterburner}_{GRPO}$ models involves two steps: first, a base model is finetuned on the $DS_{cold\_start}$ dataset to ensure adherence to the required response format; thereafter, these models are trained on $DS_{GRPO}$.

### D.2 Details of Afterburner SFT

**Training.** We fine-tune *Qwen/Qwen2.5-3B-Instruct* using Low-Rank Adaptation (LoRA). The model is trained for one epoch on $DS_{SFT}$. Key hyperparameters include a learning rate of 3e-5, managed by a cosine scheduler with 200 warm-up steps, an effective batch size of 64 (per-device batch size of 4 with 16 gradient accumulation steps), and the *adamw_torch optimizer*. For LoRA, the rank is 8 and alpha is 16. The training uses BF16 precision, and gradients are clipped at a norm of 1.0.

### D.3 Details of Afterburner DPO

**Training.** $\texttt{Afterburner}_{DPO}$ is trained from the checkpoint of $\texttt{Afterburner}_{SFT}$ utilizing LoRA for one epoch of $DS_{DPO}$ dataset. Key hyperparameters include: *learning_rate=4e-5* with a cosine scheduler and 300 warm-up steps, an effective batch size of 16 (per-device batch size of 2 with 8 gradient accumulation steps), and the *adamw* optimizer. LoRA parameters are set to rank 16, alpha 16, and a dropout of 0.05. DPO-specific settings include a beta of 0.1 and a sigmoid loss function, with pref_ftx (SFT loss component) set to 0. The training uses BF16 precision, and gradients are clipped at a norm of 1.0.

### D.4 Details of Afterburner Code Start

**Model Response Collection.** We collect the model response from *gemini-2.5-pro-exp-03-25*, using the system prompt as shown in Section E and the $\texttt{Afterburner}$ inference prompt as shown in Section E. We only keep the responses that can pass the response regex filter as shown in Section D.5.

**Training.** We conduct SFT on base model *Qwen2.5-3B-Instruct* using the LLaMA Factory framework [75]. The model undergoes full fine-tuning on an epoch of the $DS_{COLD}$ dataset, with input sequences processed up to a maximum length of 32,768 tokens. Key hyperparameters included *learning rate=5e-5*, managed by a cosine scheduler with *50 warm-up steps*, an effective batch size of 4, and the *adamw_bnb_8bit* optimizer.

### D.5 Details of Afterburner GRPO

**Training.** $\texttt{Afterburner}_{GRPO}$ is trained on Verl [58] and initialized from $\texttt{Afterburner}_{CS}$. The GRPO training runs for *20 epochs* on $DS_{GRPO}$. Since executing generated code and computing

its efficiency metrics are time-consuming, we use a batch reward function to accelerate the reward calculation in a parallel manner. Key hyperparameters include: *actor_learning_rate=1e-6*, *ppo_mini_batch_size=32* (4 per-GPU micro-batch). During the roll-outs, 16 responses are generated per prompt using vLLM [39] with *inference_temperature=1.0*. KL loss for actor updates is disabled, and the entropy coefficient is 0. For the reward weights, we set $\beta_f = 0.2, \beta_e = 0.3, \beta_c = 0.5$. Note that $\mathcal{R}_{efficiency}$ is set to 0 if $C'_{pass} = 0$. $e_{upper}$ is set to *90, 1048576, 94371840*, respectively, which aligns with our timeout (*90s*) and memory (*1GB*) limitation. In our experiment, we observed that a dominant weight for functional correctness ($\beta_c \geq 0.5$) is essential for stable training, as values below this threshold often led to training crash. To balance our objectives, we set a fixed $\beta_c = 0.5$ while gradually increasing the code efficiency weight $\beta_e$ from 0.3 to 0.5. This strategy ensures the model first learns to produce correct code before optimizing for efficiency.

**Format Regex.** Inspired by recent works [20, 57, 37, 63, 19], we encourage our model to generate the reasoning content before the code solution. The designated response format: *"<thinking> thing_content </thinking> <solution> solution_content </solution>"*.

**Afterburner Format Regex**

```python
import re
def single_thinking_solution_format(text: str) -> bool:
    pattern = re.compile(
        r"""
        \A\s* # optional leading whitespace
        <thinking>
            (?:(?!<thinking>).)*?
        </thinking>\s* # end <thinking>
        <solution>
            (?:(?!<thinking>|<solution>).)*?
        </solution>\s* # end <solution>
        \Z
        """,
        re.DOTALL | re.VERBOSE,
    )
    return bool(pattern.fullmatch(text))
```

**Reward Function Design for Enhanced Code Generation**  The efficacy of our Group Relative Policy Optimization (GRPO) framework, particularly for a task as nuanced as code generation, heavily relies on a well-designed reward function. Our objective is to guide the Afterburner$_{GRPO}$ model not merely towards syntactically valid code, but towards solutions that are functionally correct, computationally efficient, and adhere to a desired structured output format that includes an explicit reasoning phase. To this end, our final reward $\mathcal{R}_{final}$ is a carefully weighted composite of three distinct components, each targeting a critical aspect of code quality.

**Format Control ($R_{Format}$).** We first incentivize adherence to a predefined output structure, which mandates a thinking phase encapsulated in `<thinking>...</thinking>` tags followed by the code within `<solution>...</solution>` tags. As defined in Eq. (6), $R_{Format}$ provides a strong binary signal ($+1$ for compliance, $-1$ otherwise). This not only ensures predictable and parsable outputs for automated assessment but also explicitly encourages the model to engage in a "thought process" prior to generating the final solution, a step we believe is crucial for complex problem-solving.

**Functional Correctness ($R_{correct}$).** Ensuring functional soundness is paramount. However, a simple binary pass/fail reward for the current generation $C'$ can be a sparse and inefficient signal. Instead, $R_{correct}$ (Eq. (7)) evaluates $C'$ in comparison to a baseline attempt $C$. It assigns the highest positive reward ($1.0$) for an "upgrade" (i.e., $C'$ passes tests while $C$ fails) and the largest penalty ($-1.0$) for a "downgrade" ($C'$ fails while $C$ passes). Maintaining a passing or failing status yields intermediate rewards ($0.5$ and $-0.5$ respectively). This relative assessment provides a more nuanced gradient, strongly favoring improvements and robustly penalizing regressions.

**Efficiency Improvement ($\mathcal{R}_{efficiency}$).** Beyond correctness, generating efficient code is our key objective. $\mathcal{R}_{efficiency}$ (Eq. (8)) is designed to reward relative improvements in computational performance (e.g., time, memory). The core of this reward is $e_{gain}$, which measures the normalized improvement

of the current solution $C^{out}$ over a baseline $C^{in}$, after clipping efficiency metrics to a sensible range $[0, e_{upper}]$ to handle outliers. Crucially, we apply the hyperbolic tangent function ($\tanh$) to $e_{gain}$. This bounds the reward component within $(-1, 1)$, providing a smooth, scaled signal that is sensitive to gains but diminishes returns for extremely large improvements or degradations, thereby stabilizing the learning process. A small $\epsilon$ in the denominator of $e_{gain}$ ensures numerical stability. Furthermore, the inherent stochasticity often present in empirical efficiency measurements (e.g., due to minor system-level variations or non-deterministic aspects of complex code execution) means that $\mathcal{R}_{efficiency}$ naturally introduces a degree of noise. This moderate, implicit stochasticity can be beneficial for GRPO, as it helps maintain variance in reward signals across roll-outs. This, in turn, can prevent the advantage term $\mathcal{A}_i$ (Eq. (11)) from prematurely collapsing or vanishing, thereby fostering continued exploration and more robust policy updates.

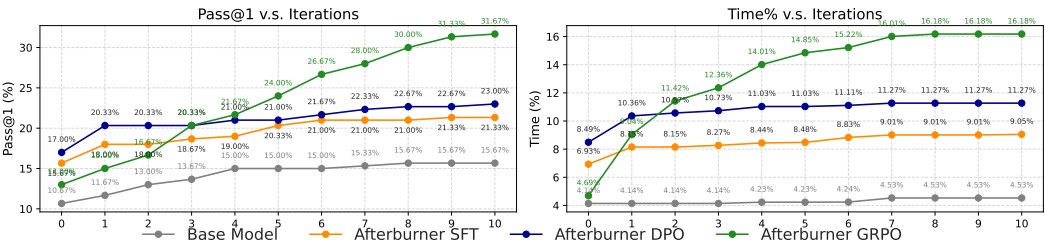

Figure 9: Iterative Optimization Performance on APPS.

# E   Model Prompts

```
                    ┌──────────────────────────────────┐
                    │  Afterburner Prompt Template     │
     ┌──────────────┴──────────────────────────────────┴────────────────┐
     │ 1 ## Instructions                                                   │
     │ 2 Your task is to implement a solution to the following problem in   │
     │       {target_lang}.                                                 │
     │ 3                                                                    │
     │ 4 ## Problem Description                                             │
     │ 5 {problem_description}                                              │
     │ 6                                                                    │
     │ 7 ## Original Solution                                               │
     │ 8 {original_solution}                                                │
     │ 9                                                                    │
     │10 ## Original Performance                                            │
     │11 Passed: {original_passed} / Time: {original_time} / Memory: {      │
     │       original_memory} / Integral: {original_integral}               │
     │12                                                                    │
     │13 ## Output Format                                                   │
     │14 - Provide the complete solution code in one markdown code block    │
     │       with an appropriate language identifier.                       │
     │15 - Generate the initial solution code directly if Original          │
     │       Solution is empty.                                             │
     │16 - Fix the original solution if it was not passed. Optimize the {   │
     │       efficiency_instruction} performance if the original solution   │
     │       was passed.                                                    │
     └────────────────────────────────────────────────────────────────────┘
```

# F   Would the History-Aware Model Perform Better?

We attribute the under-performance of history-aware model in Table 10 to two primary factors: **1) Distribution Shift:** The performance drop is likely caused by a distribution shift, as the training format (single-turn) differs from the multi-turn, history-aware format used during inference. **2) Training Challenges:** While fine-tuning the model with multi-turn format could mitigate the distribution shift, this approach introduces the significant challenge of long-term credit assignment, a well-known difficulty in Reinforcement Learning [15].

Table 10: Performance of Single-turn Loop (Afterburner) and Multi-turn Loop (History-aware).

| Iteration | 0 | 1 | 2 | 5 | 7 | 10 |
|---|---|---|---|---|---|---|
| **Single-turn Loop** | | | | | | |
| Pass % | 47.33 | 50.33 | 52.00 | 58.17 | 60.50 | 61.67 |
| Beyond-I % | 18.24 | 24.81 | 29.44 | 35.48 | 38.01 | 38.95 |
| **Multi-turn Loop (History-aware)** | | | | | | |
| Pass % | 45.00 | 46.33 | 46.33 | 46.33 | 46.67 | 46.67 |
| Beyond-I % | 15.10 | 15.10 | 15.10 | 15.10 | 15.10 | 15.10 |

The above results show that applying the history-aware method directly to a vanilla model yielded no performance improvement over the single-turn baseline. Despite instructions to avoid repetition, the model predominantly replicated existing solutions from the prompt. The few novel solutions it generated, while claiming to be superior, demonstrated no empirical performance gains. This observation aligns with our core hypothesis: *vanilla models can generate correct code but lack an intrinsic awareness of code efficiency.*

# G   Uncertainty in Code Efficiency Measurement

Quantifying code efficiency is a nuanced challenge. Theoretical efficiency, typically expressed via asymptotic notation (e.g., $O(nlogn)$), offers high-level algorithmic understanding but often neglects constant factors, compiler optimizations, and hardware-specific impacts (like cache performance or instruction-level parallelism) crucial for real-world performance. Consequently, it provides an incomplete picture for comparing concrete code implementations.

Table 11: A summary of different training and inference schemes.

| Training / Inference | Single-Turn | Multi-Turn |
|---|---|---|
| Single-Turn | **Afterburner** (P) Consistent distribution. (P) Stable RL training. (P) Avoids credit assignment. (C) Cannot handle history. | **Training Waste** (C) It wastes the model's trained ability. |
| Multi-Turn | **Distribution Mismatch** (C) Critical mismatch between training and inference data leads to poor performance. | **Ideal History-Aware** (P) Creates a truly context-aware agent. (C) Suffers from sparse rewards and unstable training. (C) Difficult long-distance credit assignment. |

On the other end of the spectrum, simulation and low-level statistics (e.g., cycle-accurate simulations, CPU performance counters) can provide extremely detailed data [60]. However, these methods often introduce substantial complexity in setup and interpretation, may have limited scope in accurately modeling all modern system intricacies, or can be overly specific to a particular hardware configuration, making generalization difficult. For our purposes, the granularity and setup overhead of such approaches outweigh their benefits.

We therefore opt for **empirical performance measurement**, directly observing execution metrics like runtime and memory usage. This approach holistically captures the interplay of algorithm, code structure, compilation, and the underlying hardware. While direct, empirical results are subject to inherent system noise and run-to-run variability. To rigorously address this and derive stable performance indicators, robust statistical techniques are indispensable, leading to our choice of bootstrapping for uncertainty quantification.

### G.1 Details of Bootstrapping Evaluation

To quantify the statistical uncertainty of our efficiency metrics, we employ a bootstrapping procedure [17]. Task-level efficiency metrics are first grouped by their respective IDs to ensure independent sampling for each task. We generate $B = 128$ bootstrap replicates. Each replicate is constructed by sampling $k = 4$ solutions for every unique problem (in our settings, we repeatedly evaluate each generated code 16 times). For each of these $B$ replicates, we then calculate the Average BEYOND-T, BEYOND-M, and BEYOND-I. Finally, we report the mean of each of these four metrics across all replicates, along with their corresponding 95% confidence intervals, to offer a robust evaluation of model performance.

## H Monolith Implementation

**Code Execution Environment.** We deploy a code execution environment on a GCP *n2-highcpu-96* instance (96 vCPUs, 96 GB Memory) with 81 `Monolith` workers. Each worker operates within a dedicated Docker container [13], which is allocated 1 vCPU, 1 GB of memory, and provided with an isolated temporary directory. To ensure a pristine execution environment for each evaluation, containers are created anew for every task. CPU affinity for each worker was set to 100% to minimize performance variability during measurements. Execution time and peak memory overhead were measured using the *'time -v'* command. To gather instantaneous memory usage and calculate the integral score, we sampled the *'VmRSS'* field from the process status file (*/proc/[pid]/status*). To accelerate model inference, we use the batch inference feature on Neibus [†] for all available models listed in Table 4. For proprietary models, we call their provided APIs. For those models without an online inference point, we host vLLM [39] inference service locally. Further details on the execution environment are available in the Appendix H.

**Runtime.** The monolith's runtime environment is standardized using Docker containerization to ensure consistency and portability across different programming languages. Each language or service

---

[†] https://studio.nebius.com/

within the monolith operates within a specific, pre-defined Docker image. Table 12 details the official Docker images utilized for various supported programming languages.

Table 12: Programming Language Docker Images

| Language | Image |
|----------|-------|
| Python | python:3.9.19-bullseye |
| Java | openjdk:11.0.12-jdk-bullseye |
| Javascript | node:22-bullseye |
| Cpp | gcc:11.2.0-bullseye |
| Go | golang:1.17.0-bullseye |
| Ruby | ruby:3.0.2-bullseye |
| Rust | rust:1.85.0-bullseye |

# I  Symbol List

| Symbol / Term | Description |
|---------------|-------------|
| LLM | Large Language Model |
| SFT | Supervised Fine-Tuning |
| DPO | Direct Preference Optimization |
| GRPO | Group Relative Policy Optimization |
| RL | Reinforcement Learning |
| IOF | Iterative Optimization Framework (the proposed framework) |
| Afterburner | Code optimization models (trained via SFT, DPO, GRPO) |
| Monolith | A high-fidelity code execution sandbox for performance feedback |
| Venus | A dataset with human solutions, curated for efficiency benchmarking |
| APPS | An existing dataset for code generation, also used for evaluation |
| $DS_{SFT}$ | Dataset constructed for Supervised Fine-Tuning |
| $DS_{DPO}$ | Preference dataset constructed for Direct Preference Optimization |
| $DS_{CS}$ | Cold Start Dataset used for initial format alignment of GRPO models |
| $DS_{GRPO}$ | Dataset used for Group Relative Policy Optimization training |
| $\mathcal{P}$ | Problem description |
| $\mathcal{I}$ | Efficiency instruction |
| $\mathcal{C}$ | A code solution. Variants: |
| $\mathcal{C}_i^{in}$ | Input code solution for iteration $i$ |
| $\mathcal{C}_i^{out}$ | Output (improved) code solution from Afterburner at iteration $i$ |
| $\mathcal{C}^+$ | A more efficient/preferred code solution |
| $\mathcal{C}^-$ | A less efficient/dis-preferred code solution |
| $\mathcal{C}^{baseline}$ | A baseline code solution for comparison (in DPO context) |
| $\mathcal{M}$ | Performance metric(s) of a solution. Variants: |
| $\mathcal{M}_i^{in}$ | Performance metrics of $\mathcal{C}_i^{in}$ |
| $\mathcal{M}_i^{out}$ | Performance metrics of $\mathcal{C}_i^{out}$ |
| $\succ$ | Relation indicating superior performance (e.g., $\mathcal{M}_i^{out} \succ \mathcal{M}_i^{in}$) |
| $N_{iter}$ | Total number of optimization iterations |
| $\mathcal{X}$ | Input prompt to a model (often includes $\mathcal{P}, \mathcal{I}, \mathcal{C}^{in}, \mathcal{M}^{in}$) |
| $\pi_\theta$ | The policy (language model) being trained, parameterized by $\theta$ |
| $\pi_{ref}$ | Reference policy (e.g., in DPO, the SFT model) |
| $\mathcal{L}_{SFT}$ | Loss function for Supervised Fine-Tuning |
| $\mathcal{L}_{DPO}$ | Loss function for Direct Preference Optimization |
| $\mathcal{L}_{GRPO}$ | Loss function for Group Relative Policy Optimization |
| $\beta$ | Weights for reward components ($\beta_f, \beta_c, \beta_e$). |
| $\sigma(\cdot)$ | The logistic function |

*Continued on next page*

| Symbol / Term | Description |
|---|---|
| $R_{Format}$ | Reward component for adhering to the specified output format |
| $R_{correct}$ | Reward component for functional correctness |
| $\mathcal{R}_{efficiency}$ | Reward component for improvement in computational efficiency |
| $\mathcal{E}$ | An absolute code efficiency value (e.g., execution time, peak memory) |
| $\mathcal{E}_{gain}$ | Normalized relative gain in an efficiency metric $e$ |
| $\mathcal{E}_{clip}$ | Clipped value of an efficiency metric $e$ |
| $\mathcal{E}_{upper}$ | Upper limit for clipping an efficiency metric $e$ |
| $\mathcal{R}_{final}$ | The final combined reward signal for GRPO |
| $\mathcal{O}_i$ | The $i$-th generated output (rollout/candidate solution) in a GRPO group |
| $G$ | Size of the rollout group in GRPO |
| $\mathcal{W}_i$ | Policy ratio (importance weight) for rollout $\mathcal{O}_i$ in GRPO, $\frac{\pi_\theta(\mathcal{O}_i\|\mathcal{X})}{\pi_{\theta_{\text{old}}}(\mathcal{O}_i\|\mathcal{X})}$ |
| $\mathcal{A}_i$ | Advantage of rollout $\mathcal{O}_i$ within its group in GRPO |
| $\epsilon$ | A small constant |
| PASS@1 | Percentage of the first generated solution passes all test cases |
| $PR(x, D)$ | Percentile Rank: fraction of items in distribution $D$ that $x$ is greater than or equal to (for efficiency, lower is better, so $1 - PR$ or adjusted PR is used implicitly if higher means better) |
| BEYOND-T | Global efficiency metric: average percentile rank of generated code's execution time relative to human solutions. Higher is better. |
| BEYOND-M | Global efficiency metric: average percentile rank of generated code's memory usage relative to human solutions. Higher is better. |
| BEYOND-I | Global efficiency metric: average percentile rank of generated code's integral score relative to human solutions. Higher is better. |
| $r_k^{gen}, m_k^{gen}, i_k^{gen}$ | Absolute execution time, peak memory usage, and integral score of the generated solution for task $k$ |
| $D_k^T, D_k^M, D_k^I$ | Distributions of execution times, memory usages, and integral scores from reference human solutions for task $k$ |
| B% | Percentage of generations that are better than all human solutions |
| M% | Percentage of generations whose efficiency falls within the range of human solutions |
| W% | Percentage of generations that are worse than all human solutions |
| F% | Percentage of failed model generation |

