# OpenReview forum: "Afterburner: Reinforcement Learning Facilitates Self-Improving Code Efficiency Optimization"
_NeurIPS.cc/2025/Conference — NeurIPS 2025 poster_

### Official Review · Reviewer_UfKJ · 2025-07-01

**Clarity:** 3
**Significance:** 2
**Originality:** 2
**Rating:** 4
**Confidence:** 4

**Summary:**

This paper introduces an Iterative Optimization Framework (IOF) for improving code efficiency at test time using feedback-driven refinement. In each iteration, a language model receives the current version of the code and execution-based feedback (e.g., peak memory usage, latency, total memory) from a sandboxed runtime environment, then generates an improved version of the code. The authors evaluate various training strategies—Supervised Fine-Tuning (SFT), Direct Preference Optimization (DPO), and Group Relative Policy Optimization (GRPO)—for their ability to adapt under the IOF setting. They further release Venus, a new dataset consisting of programming problems and associated solution variants with varying efficiency profiles.

**Questions:**

- See Weaknesses section
- How does IOF performance vary when initialized with code that spans the efficiency spectrum (from highly inefficient to highly efficient)? Are GRPO-trained models more robust to such variation?
- Have you conducted ablations on the number of iterations (e.g., convergence rate) or observed performance ceilings?

**Ethical Concerns:**

["NO or VERY MINOR ethics concerns only"]

**Final Justification:**

The authors have addresses all my concerns with adequate experiments and justification. While using iterative feedback to improve LLM responses is not completely novel, using it for improving code efficiency is an interesting use case. Thus I am increasing my score from Reject to Borderline Accept.

**Limitations:**

Yes, the authors have discussed their Limitations but I would suggest it to move it from the Appendix to the Main Body of the paper.

**Quality:**

2

**Strengths And Weaknesses:**

**Strengths**

Clear Framework Design: The paper presents a well-defined framework for feedback-driven code-efficiency refinement. The iterative nature of the process and the way it leverages runtime metrics are clearly explained.

Clear Dataset Description: The Venus dataset and associated training data are systematically constructed and properly explained. This is a valuable resource for the community focused on efficiency-aware code generation.

Empirical Gains from GRPO: The GRPO-trained Qwen2.5 3B model demonstrates significantly better test-time efficiency improvements over baseline and SFT/DPO-trained models within the IOF, showcasing the method's potential.

**Weaknesses**

1. Incomplete Evaluation of IOF Design Choices:
The current implementation generates improved code based only on the most recent iteration. Would a history-aware model  (i.e., $C_{0:i-1}, M_{0:i-1}$) perform better? Using all previous iterations could be useful to break out of local minima where $M_{i}^{out}=M_i^{in}$ despite code differences $C_{i}^{out} != C_i^{in}$?

2. Lack of Scaling Studies:
All key results focus on the Qwen2.5 3B model. It's unclear whether GRPO and IOF yield consistent improvements for larger-scale models (e.g., 7B, 13B). This is critical to assess the generalizability and practical relevance of the framework.

3. Limited Baseline Comparisons in IOF Iterations:
Table 1 compares performance across different models only at iteration 0, but does not show how baseline models (e.g., vanilla GPT, LLama4, Claude, etc.) evolve under IOF. This obscures how much of the final performance gain comes from model training (e.g., GRPO) versus the framework itself. Showing relative improvement over iterations (e.g. $C_0$ against $C_{i_{best}}$) for different base models would help isolate the benefits.

4. Insufficient Justification for Venus Dataset:
The paper lacks a clear comparison with existing datasets such as Mercury, EffiBench, and EvalPerf. What are the limitations of those datasets that Venus addresses? What is the statistical diversity (e.g., memory, time distributions) of reference solution variants in Venus compared to prior work?

5. Missing Benchmark Analysis on Existing Datasets:
There is no empirical evaluation of IOF and GRPO-trained models on established efficiency-focused code benchmarks like Mercury, EffiBench. This limits the ability to compare against prior art and undermines claims of generalizability.

---

> ### Author Rebuttal · Authors · 2025-07-31
>
> Dear Reviewer UfKJ,
>
> Thank you for your constructive and insightful reviews. We hope we can address your concerns as follows:
>
> > Q1: Would a history-aware model perform better?
>
> We conduct the experiment as you suggested, and show results in the following table. The results below show that the history-aware model did not outperform the single-turn model in the current setting.
>
> | Single-turn Loop Iteration | 0 | 1 | 2 | 5 | 7 | 10 |
> |---|:---:|:---:|:---:|:---:|:---:|:---:|
> | Pass %  | 47.33 | 50.33 | 52.00 | 58.17 | 60.50 | 61.67 |
> | Beyond-I %  | 18.24 | 24.81 | 29.44 | 35.48 | 38.01 | 38.95 |
>
> | Multi-turn Loop (History-aware) Iteration | 0 | 1 | 2 | 5 | 7 | 10 |
> |---|:---:|:---:|:---:|:---:|:---:|:---:|
> | Pass %  | 45.00 | 46.33 | 46.33 | 46.33 | 46.67 | 46.67 |
> | Beyond-I %  | 15.10 | 15.10 | 15.10 | 15.10 | 15.10 | 15.10 |
>
> We attribute the underperformance of history-aware model to two primary factors: **1) Distribution Shift:** The performance drop is likely caused by a distribution shift, as the training format (single-turn) differs from the multi-turn, history-aware format used during inference. **2) Training Challenges:** While fine-tuning the model with multi-turn format could mitigate the distribution shift, this approach introduces the significant challenge of long-term credit assignment, a well-known difficulty in Reinforcement Learning. We agree that history-aware modeling is a valuable and promising research direction. We appreciate the insightful feedback and plan to explore this more thoroughly in our future work.
>
> > Q2: Model Scale Generalization
>
> While this work focuses on exploring the methodology of iterative efficient optimization rather than the scalability of larger models, we do agree that it is critical to evaluate our frame on larger models. Due to the rebuttal time limitation, we conducted additional experiments on Qwen2.5 7B as you suggested. The result demonstrates our iterative optimization framework can be generalized to larger models.
>
> | **Iteration** | 0 | 1 | 2 | 5 | 7 | 10 |
> |---|:---:|:---:|:---:|:---:|:---:|:---:|
> | **Pass %** | 60.67 | 65.67 | 69.33 | 74.67 | 79.67 | 83.67 |
> | **Beyond-T %** | 27.67 | 32.87 | 34.22 | 39.50 | 41.91 | 42.86 |
> | **Beyond-M %** | 29.79 | 31.55 | 36.48 | 40.97 | 43.26 | 44.51 |
> | **Beyond-I %** | 21.02 | 28.10 | 33.85 | 37.54 | 37.54 | 38.66 |
>
> > Q3: Baseline Comparisons in IOF Iterations
>
> We thank the reviewer for this insightful suggestion. To better isolate the benefits of our framework versus our model training approach, we have conducted new experiments applying IOF to several baseline models as requested. Specifically, We ran the IOF process for 8 iterations on *the vanilla Qwen-2.5-3B, Qwen-2.5-7B, and OpenAI GPT-4o models*. The results are presented below:
>
> | **Qwen2.5 3B Iterations** | 0 | 1 | 2 | 4 | 8 |
> |:---:|:---:|:---:|:---:|:---:|:---:|
> | **Pass %** | 27.99 | 28.33 | 28.67 | 29.00 | 29.33 |
> | **Beyond-T %** | 12.40 | 12.74 | 13.08 | 13.42 | 13.69 |
> | **Beyond-M %** | 13.24 | 13.73 | 14.22 | 14.71 | 15.19 |
> | **Beyond-I %** | 10.29 | 10.77 | 11.25 | 11.72 | 12.07 |
>
> | **Qwen2.5 7B Code Iterations** | 0 | 1 | 2 | 4 | 8 |
> |:---:|:---:|:---:|:---:|:---:|:---:|
> | **Pass %** | 52.21 | 54.00 | 54.33 | 56.00 | 58.33 |
> | **Beyond-T %** | 20.66 | 21.47 | 23.03 | 23.47 | 26.26 |
> | **Beyond-M %** | 25.21 | 25.72 | 26.48 | 29.93 | 31.20 |
> | **Beyond-I %** | 16.78 | 17.69 | 20.91 | 23.21 | 27.14 |
>
> | **GPT-4o Iterations** | 0 | 1 | 2 | 4 | 8 |
> |---|:---:|:---:|:---:|:---:|:---:|
> | **Pass %** | 82.26 | 84.00 | 84.33 | 84.33 | 85.33 |
> | **Beyond-T %** | 38.22 | 40.16 | 42.02 | 44.16 | 46.17 |
> | **Beyond-M %** | 42.09 | 43.28 | 44.71 | 46.20 | 48.01 |
> | **Beyond-I %** | 28.89 | 30.60 | 32.55 | 35.02 | 41.29 |
>
> As the data shows, larger models do achieve better performance through the IOF iterations. However, the improvement in these vanilla models is significantly less than that of our Afterburner models. This demonstrates that while the IOF framework itself is beneficial, our proposed training paradigm are the primary drivers of the substantial performance gains we report.
> (Note: As closed-source models have not disclosed their training details for, we cannot determine if similar reinforcement learning techniques were already incorporated into their development pipeline.)
>
> > Q4: Venus Justification
>
> We recognize the importance of a clear comparison with prior work and will clarify this in the revised paper.
> - **Multilingual Coverage:** A primary contribution is that Venus is the first multilingual code efficiency benchmark, covering six languages. Prior work, including Mercury [1], EffiBench [2], and EvalPerf [3], focuses exclusively on Python.
> - **Solutions Diversity:** Venus offers a significantly larger and more diverse set of reference solutions. While EffiBench uses a single baseline and Mercury averages 18.4 solutions per problem, Venus provides an average of 79.3 solutions per problem for each language. This massive increase directly contributes to the statistical diversity of performance metrics, as shown in the time and memory distributions in Appendix Figure 7.
> - **Evaluation Dimensions:** As a direct extension of Mercury, which only measures execution time, Venus evaluates execution time, memory usage, and their integral, providing a more holistic assessment of code efficiency.
>
> We will incorporate this detailed comparison, currently summarized in Appendix C, into the main body to make the justification for Venus more clear.
>
> > Q6: IOF Efficiency Spectrum
>
> It's a good suggestion! We conduct the experiment to start Afterburner SFT and Afterburner GRPO with the efficiency spectrum. Since we are not allowed to upload images in the rebuttal response, we present iterative Beyond-I results beginning from the inefficient (0% Beyond-I) to the efficient (50% Beyond-I) solutions in the following table. I will update the complete efficiency spectrum figure in the revision.
>
> | **Less Efficient Start Iterations** | 0 | 1 | 2 | 4 | 8 |
> |---|---:|---:|---:|---:|---:|
> | Afterburner SFT | 3.76 | 4.74 | 5.24 | 7.31 | 11.50 |
> | Afterburner GRPO | 3.76 | 4.28 | 5.21 | 8.60 | 13.85 |
> | **More Efficient Start Iterations** | 0 | 1 | 2 | 4 | 8 |
> | Afterburner SFT | 51.44 | 51.44 | 52.93 | 54.21 | 56.73 |
> | Afterburner GRPO | 51.44 | 53.70 | 55.12 | 58.21 | 62.17 |
>
> We argue that a key advantage of RL is exploring more sufficiently in the exploitation space than SFT or DPO. Therefore, it is more robust if the inference distribution aligns with the training distribution.
>
> > Q7: Performance Ceilings
>
> Yes, In our experiments, performance ceilings are observed from IOF round 7 on both the Venus and APPS benchmarks, as shown in Figures 5 and 9. We hypothesize that these ceilings result from the trade-off between task complexity and the model capability.
> We conduct an additional experiment on three difficulty categories. For a model with a fixed capability, more difficult tasks require more iterations to converge on an optimal solution, whereas on simpler tasks, performance saturates and hits this ceiling much earlier. The following table shows that the iteration of performance ceilings (less than 5% efficiency gap with the final iteration) on Venus and APPS:
>
> |  | **Easy** | **Medium** | **Hard** |
> |---|:---:|:---:|:---:|
> | **Venus (Beyond-T)** | 3 | 6 | 7 |
> | **APPS (Beyond-T)** | 4 | 5 | 6 |
> | **Venus (Beyond-M)** | 3 | 5 | 8 |
> | **APPS (Beyond-M)** | 2 | 4 | 9 |
> | **Venus (Beyond-I)** | 4 | 5 | 8 |
> | **APPS (Beyond-I)** | 3 | 5 | 9 |
>
> > L1: Limitation Location
>
> Thank you for your suggestion. We will move it from the appendix to the main body.
>
> **We hope our explanation can address your concern. Please ask follow-ups if you have any other questions. Thank you!**
>
> [1] Mercury: A code efficiency benchmark for code large language models.
>
> [2] Effibench: Benchmarking the efficiency of automatically generated code.
>
> [3] Evaluating language models for efficient code generation.

---

> ### Comment · Reviewer_UfKJ · 2025-08-06
>
> I thank the authors for taking the time to conduct additional experiments to address my concerns. However, I still have remaining reservations regarding Q1 Q5 and Q6, which I elaborate on below:
>
> > Q1: Would a history-aware model/framework perform better?
>
> It remains unclear whether a history-aware model would yield better performance. The current experiment evaluates a single-turn trained model in a multi-turn setting, introducing distribution shift and making the results difficult to interpret. A more direct experiment would involve explicitly including the history of code improvements and corresponding efficiency feedback from the iterative optimization trace in the prompt. Specifically, to predict $C_i$, the model could be conditioned on ${C_0,M_0,…, C_{i−1}, M_{i−1}$.
>
> A comparison between this history-aware design and the current approach could yield important insights into the potential benefits of leveraging longer optimization traces—insights that are currently missing from the paper.
>
> > Q5: Missing Benchmark Analysis on existing Datasets like Mercury and EffiBench
>
> > Q6: IOF Efficiency Spectrum
>
> While the additional results show efficiency gains for both less and more efficient initializations, it remains unclear how far these improvements can be pushed with additional iterations. Notably, the last two columns indicate there is still substantial headroom. It would be helpful to understand whether starting from a less efficient code leads the IOF toward a global minimum or if the model tends to get stuck in local optima. Exploring this behavior over extended optimization horizons would help contextualize the limits and strengths of the proposed framework.

---

> > ### Author Response · Authors · 2025-08-08
> > **Further Responses [Part 1/2]**
> >
> > Dear Reviewer UfKJ,
> >
> > Thank you for your insightful follow-up questions. We are pleased that our previous response resolved most of your concerns, and we appreciate the opportunity to provide further clarification.
> >
> > > Q1: Would a history-aware model/framework perform better?
> >
> > **Our previous experiment has confirmed that the history-aware model would not perform better on single-turn trained models.** As you suggested, we managed to conduct additional experiments on exploring the performance of the vanilla model. We follow exactly your suggested format that explicitly includes the history of code improvements and corresponding efficiency feedback from the iterative optimization trace in the prompt:
> >
> > | Input | Output |
> > |---|---|
> > | (None, None) | C_0 |
> > | (C_0, M_0) | C_1 |
> > | (C_0, M_0, C_1, M_1) | C_2 |
> > | ... | ... |
> > | (C_0, M_0, C_1, M_1 … C_{i-1}, M_{i-1}) | C_{i} |
> >
> > ### Results of the Vanilla Model on the history-aware framework
> >
> > | **Iterations** | 0 | 1 | 2 | 4 | 8 |
> > |:---:|:---:|:---:|:---:|:---:|:---:|
> > | **Pass %** | 28.33 | 28.33 | 28.33 | 28.67 | 28.67 |
> > | **Beyond-T %** | 12.45 | 12.61 | 12.91 | 13.14 | 13.19 |
> > | **Beyond-M %** | 13.19 | 13.50 | 14.12 | 14.27 | 14.82 |
> > | **Beyond-I %** | 10.28 | 10.64 | 10.95 | 11.22 | 11.87 |
> >
> > The above results show that **applying the history-aware method directly to a vanilla model yielded no performance improvement over the single-turn baseline.** Despite instructions to avoid repetition, the model predominantly replicated existing solutions from the prompt. The few novel solutions it generated, while claiming to be superior, demonstrated no empirical performance gains. This observation aligns with our core hypothesis: *vanilla models can generate correct code but lack an intrinsic awareness of code efficiency.*
> >
> > |  | **Inference: Single-Turn** | **Inference: Multi-Turn** |
> > |---|---|---|
> > | **Training:Single-Turn** | [Afterburner]  - (P) Consistent data distribution. - (P) Stable RL training. - (P) Avoids credit assignment issues. - (C) Cannot handle history. | [Distribution Mismatch] - (C) Critical mismatch between training and inference data leads to poor performance. |
> > | **Training:Multi-Turn** | [Training Waste] - (C) It wastes the model's trained ability. | [Ideal History-Aware] - (P) Creates a truly context-aware agent. - (C) Suffers from sparse rewards and unstable RL training. - (C) Difficult long-distance credit assignment. |
> >
> > In conclusion, we recognize that **the history-aware framework (with longer optimization traces) can be a very promising direction, while it has a significant difference in focus from this paper**. We focus on the single-turn self-improving framework in this work to ensure consistent input distributions and stable RL training. We'd like to explore this history-aware framework in future works.
> >
> > > Q5: Missing Benchmark Analysis on existing Datasets like Mercury and EffiBench
> >
> > Sorry for missing this point in our previous response. Since Venus is a direct extension of Mercury and EffiBench, it covers all tasks in Mercury and EffiBench.
> > which is the reason why we initially focused our analysis on Venus. As requested, we have now conducted experiments on the Mercury benchmark. The results are consistent with our findings on Venus and further validate the effectiveness of our methods.
> >
> > ### Afterburner-SFT on Mercury
> >
> > | **Iteration** | **0** | **1** | **2** | **4** | **8** |
> > |---|---|---|---|---|---|
> > | Pass | 58.98 | 58.98 | 59.77 | 60.55 | 60.55 |
> > | Beyond | 26.28 | 26.95 | 27.43 | 29.65 | 31.42 |
> >
> > ### Afterburner-GRPO on Mercury
> >
> > | **Iteration** | **0** | **1** | **2** | **4** | **8** |
> > |---|---|---|---|---|---|
> > | Pass | 57.42 | 59.77 | 61.72 | 67.19 | 69.14 |
> > | Beyond | 25.84 | 27.41 | 29.5 | 32.69 | 37.23 |
> >
> > We will add these results and a corresponding discussion to the revision.

---

> > > ### Author Response · Authors · 2025-08-08
> > > **Further response [Part 2/2]**
> > >
> > > > Q6: IOF Efficiency Spectrum
> > >
> > > Thank you for your constructive feedback. We believe adding the efficiency spectrum will be very beneficial for our work. Due to the rebuttal's length constraints, we were unable to provide the complete iteration results previously. As shown in our previous response to Q7, efficiency performance for most tasks begins to converge around iteration 5 (note that the last two columns stand for iteration 4 and 8 in our previous response). More difficult tasks require additional iterations to reach their performance ceiling.
> > >
> > > Following your suggestion, we manage to conduct an additional experiment to **explore if starting from a less efficient solution traps the model in a local optimum**. We compared our Afterburner-GRPO against the Afterburner-SFT baseline.
> > >
> > > | **Iteration**    | **0** | **8** | **12** | **16** | **20** | **24** | **28** | **29** | **30** | **31** | **32** |
> > > |------------------|------:|------:|-------:|-------:|-------:|-------:|-------:|-------:|-------:|-------:|-------:|
> > > | Afterburner SFT  |  3.76 | 10.49 |  11.50 |  11.50 |  11.87 |  12.04 |  12.04 |  12.04 |  12.04 |  12.04 |  12.04 |
> > > | Afterburner GRPO |  3.75 | 14.15 |  16.00 |  20.42 |  25.59 |  27.05 |  27.05 |  28.14 |  29.01 |  29.01 |  29.13 |
> > >
> > > When initialized from a less efficient solution, Afterburner-SFT stagnates in a local optimum by iteration 24. In contrast, Afterburner-GRPO proves significantly more robust, continuing to improve and achieving a much higher performance level. This outcome highlights the superiority of our RL-based approach in exploring the solution space more effectively to escape such suboptimal traps.
> > >
> > > **We hope the results and explanation can address your concern. If you have any other questions, we are happy to provide further clarification needed.**

---

> > > > ### Comment · Reviewer_UfKJ · 2025-08-08
> > > >
> > > > Thanks for further clarifying my concerns. It is interesting to observe that history aware models even when trained with it is surprisingly doing poor as compared to single turn. Maybe the issues are worth investigating for future work.
> > > >
> > > > Furthermore, given a programming problem, starting from a less efficient solution never reaches the final convergence point of the instance which starts with a more efficient solution to the problem. This suggests there exists some global minima but not every starting code could lead to it and can get trapped in local minima (29.13 v/s 62.17). An interesting observation worth investigating where techniques from the field of optimisation could be of help. Nonetheless the proposed GRPO based training indeed gives good performance as compared to baselines.
> > > >
> > > > I am more or less satisfied with author’s rebuttal and have correspondingly increased my score.
> > > >
> > > > Once again thanks for taking time to conduct more experiments.

---

> > > > > ### Author Response · Authors · 2025-08-09
> > > > > **Thank you for your positive feedback!**
> > > > >
> > > > > Dear Reviewer UfKJ,
> > > > >
> > > > > Thank you for your insightful feedback and for raising your score! We are delighted that our responses and additional experiments have addressed your concerns.
> > > > >
> > > > > We will incoperate our discussion in the revised manuscript, including the exploration of the history-aware framework, Venus dataset justification, model scaling generalization, baseline in IOF iterations, and the IOF Efficiency Spectrum.
> > > > >
> > > > > We appreaciate your time and constructive engagement, which have significantly strengthen our work.
> > > > >
> > > > > Best regards,
> > > > >
> > > > > The Authors

---

### Official Review · Reviewer_z3gd · 2025-07-02

**Clarity:** 3
**Significance:** 2
**Originality:** 2
**Rating:** 5
**Confidence:** 5

**Summary:**

The paper introduced AfterBurner, an iterative optimization framework to address the problem of LLM failing to generate efficient code despite their functional correctness. AfterBurner forms a closed loop: an LLM proposes an updated solution, the Monolith sandbox runs it and returns the runtime, memory and time measurements, and the best candidate becomes the seed for the next round. The authors explored three strategies:  Supervised Fine-Tuning (SFT), Direct Preference Optimization (DPO), and a reinforcement-learning variant Group Relative Policy Optimisation (GRPO) that leverages Monolith feedback as reward. To evaluate efficiency, the authors built Venus, a new dataset designed for rigorous code efficiency assessment, with 2181 training and 300 test tasks. The metrics they used is BEYOND-T/M/I, measuring percentile rank of runtime, peak memory and their integral against the human distribution. Experiments show that on Venus, GRPO improves pass@1 from 47% to 62%, BEYOND-I from 31% to 45%. Ablation shows that execution feedback and oracle context are essential.

**Questions:**

1. How can the proposed strategy handle API-dependent real-world code? For example, does the proposed method still work in software engineering benchmarks like SWE-bench or Defects4J? I noted this was explained in the Limitations section, but I'm still curious about the authors thoughts. It's okay to leave this unaddressed for this paper, but then I will also keep my concerns about the relatively narrowed scope.
2. The final reward has a combination of hyperparameters. How sensitive is it when given different values?
3. Could you elaborate on how you measure the statistical significance of your results?
4. Does the same conclusion hold when APPS is changed to LiveCodeBench with no problem contamination?

If these questions are properly addressed, I'm happy to raise the score.

**Ethical Concerns:**

["NO or VERY MINOR ethics concerns only"]

**Final Justification:**

The paper has made solid contributions and experiments. While my concern about its generalizability to open-ended software tasks is not addressed, the authors provide clear explanations and have addressed most of my other concerns. I'm giving an Accept.

**Limitations:**

yes

**Paper Formatting Concerns:**

Some typos, no major concerns.

**Quality:**

2

**Strengths And Weaknesses:**

### Strengths

* The experimental design is solid. It compares three optimisation paradigm and provides ablations and confidence intervals. The BEYOND metrics can mitigate some hardware variance.
* The reward design and overall technique flow is well presented.
* The paper tackles an under-explored aspect in code generation, i.e. efficiency. It matters a lot for production deployment. It also introduces a sizeable benchmark that others can reuse.
* Combining RL with execution-based rewards for efficiency rather than correctness is a novel angle

### Weaknesses

* There are many reported improvements, but their statistical significances are unknown.
* Some typos like "paassed" in line 135.
* There are no decontamination processes mentioned in the paper. APPS is also not a "live" benchmark in comparison to LiveCodeBench.
* The novelty mainly lies in the application of RL to code efficiency. GRPO itself extends as an existing algorithm.
* The focusing on just python and file-level coding makes the scope of the paper narrow. In practice, developers care more about real-world code living in real software.

---

> ### Author Rebuttal · Authors · 2025-07-31
>
> Dear Reviewer z3gd:
>
> Thank you for your constructive and insightful reviews! We hope we can address your concerns as follows:
>
> > Q1: Scaling to API-dependent Real-world Code
>
> Thank you for this insightful question. While this paper focuses on establishing a robust method for function-level optimization, we agree that scaling to repository-level benchmarks like SWE-bench is a critical next step. We view our current work as the foundational layer for this larger goal and introduce the following strategy to address the main challenges:
>
> - **API Dependencies:** To handle API-dependent code, we may integrate Retrieval-Augmented Generation (RAG) and static analysis. This allows our model to dynamically pull in relevant API definitions, usage patterns, and code snippets from the broader repository, providing the necessary context for effective optimization.
>
> - **Context Management:** Real-world repositories are too large for current model context windows. We could employ a divide-and-conquer approach. This involves identifying and isolating the most relevant functions and modules for optimization, making the task computationally tractable.
>
> - **Test Generation:** Verifying changes across an entire repository is notoriously difficult. Our strategy would focus on generating targeted unit tests specifically for the modules undergoing optimization. This ensures the correctness and efficiency of our changes in a localized, verifiable, and recursive manner.
>
> In summary, while scaling our method to repo-level presents a significant research challenge, we hope that structured approach provides a clear path forward.
>
> > Q2: Hyperparameters Sensitivity
>
> In our experiment, we observed that a dominant weight for functional correctness (beta_c​ ≥ 0.5) is essential for stable training, as values below this threshold often led to training crash. To balance our objectives, we set a fixed beta_c ​​= 0.5 while gradually increasing the code efficiency weight (beta_e) from 0.3 to 0.5. This strategy ensures the model first learns to produce correct code before optimizing for efficiency.
>
> > Q3: Statistical Significance
>
> We employ bootstrap sampling to construct 95% confidence intervals for our efficiency metrics. Here is the detailed procedure:
> 1. **Data Collection:** For each individual problem, we generate 4 unique solutions (temperature=1) and measure the performance of each one 16 times. This results in a performance set of 64 measurements (4 solutions×16 measurements).
> 2. **Bootstrap Sampling:** We then create 128 bootstrap samples. Each sample is formed by drawing one measurement with replacement from the original performance set.
> 3. **Distribution Generation:** We generate an empirical sampling distribution for each efficiency metric by aggregating the 128 bootstrap samples over all problems.
> 4. **Score Reporting:** We report the mean of these bootstrap distributions along with their 95% confidence interval.
> This method allows for a robust estimation of efficiency performance and its statistical uncertainty. We provide a detailed explanation in Appendix G.
>
> > Q4: Contamination-free Evaluation
>
> We evaluate our methods on LiveCodeBench and report the pass rate below.
> The results confirm that our conclusions hold. Afterburner-GRPO shows substantial and consistent improvement as the number of iterations increases, while Afterburner-SFT plateaus quickly. This trend is consistent with our findings on the APPS and Venus benchmarks, demonstrating the general effectiveness of our approach.
>
> | **Afterburner-SFT Iteration** | 0 | 1 | 2 | 5 | 7 | 10 |
> |---|:---:|:---:|:---:|:---:|:---:|:---:|
> | **Pass %** | 5.49 | 6.59 | 6.59 | 7.14 | 7.69 | 7.69 |
>
> | **Afterburner-GRPO Iteration** | 0 | 1 | 2 | 5 | 7 | 10 |
> |---|:---:|:---:|:---:|:---:|:---:|:---:|
> | **Pass %** | 5.49 | 7.69 | 9.89 | 11.54 | 12.09 | 13.19 |
>
> > Q5: Some typos like "paassed" in line 135.
>
> We will fix this typo in the revision.
>
> **We hope our explanation can address your concern. Please ask follow-ups if you have any other questions. Thank you!**

---

> > ### Comment · Reviewer_z3gd · 2025-08-03
> >
> > Thanks for the rebuttal that properly addressed Q3 to Q5. While my concern about its scalability to real-world code still exists, the additional explanation is helpful. I remain very positive about the work and have increased Quality and Significance scores. Before concluding on the final rating, I will keep a closer eye on the discussion from the other reviewers (especially those who gave a negative scores).
> >
> > > gradually increasing the code efficiency weight (beta_e) from 0.3 to 0.5
> >
> > You mentioned in Q2 that you gradually increased the weight. Could you elaborate on the process? Did you stop the run and continue with a different weight, or is it dynamically changed in a single training run, and how is it adjusted (say linearly)? If it's the former, how did you handle learning rate and optimizer states?

---

> > > ### Author Response · Authors · 2025-08-04
> > > **Thank you for your positive feedback!**
> > >
> > > We are glad our response addressed most of your concerns and thank you for the positive feedback!
> > >
> > > In terms of annealing schedule for code efficiency weight (beta_e), **we linearly increase beta_e within a single training run**, much like a learning rate warmup schedule. Specifically, for each training step, we calculate the current beta_e based on the training progress and feed it into our custom reward function. This strategy enables our model to first focus on functional correctness and then increasingly on code efficiency as training advances.
> > >
> > > ```python
> > > def get_reward_coefficient(current_step, total_steps, initial_coef=0.3, final_coef=0.5):
> > >     """Linearly interpolates the reward coefficient based on training progress."""
> > >     progress = current_step / total_steps
> > >     return initial_coef + (final_coef - initial_coef) * progress
> > > ```

---

> > > > ### Comment · Reviewer_z3gd · 2025-08-05
> > > >
> > > > Thanks for the clarification. I have finalized a score of 5.

---

> > > > > ### Author Response · Authors · 2025-08-09
> > > > > **Thank you for your positive feedback!**
> > > > >
> > > > > Dear Reviewer z3gd,
> > > > >
> > > > > Thank you for your positive feedback! We will incorporate our discussion into the revised manuscript, including *scalability to real-world code, hyperparameter sensitivity, statistical significance, and the contamination-free evaluation.*
> > > > >
> > > > > We appreciate the time and consideration you dedicated to our work.
> > > > >
> > > > > Best regards,
> > > > >
> > > > > The Authors

---

### Official Review · Reviewer_xncp · 2025-07-03

**Clarity:** 3
**Significance:** 3
**Originality:** 3
**Rating:** 5
**Confidence:** 4

**Summary:**

This paper introduces Afterburner, an iterative optimization framework designed to improve the computational efficiency of LLM-generated code. The framework employs a closed-loop system where LLMs iteratively refine code based on empirical performance feedback from "Monolith," a high-fidelity execution sandbox that provides real-time efficiency metrics. The authors compare three training strategies: Supervised Fine-Tuning (SFT), Direct Preference Optimization (DPO), and Group Relative Policy Optimization (GRPO), finding that while SFT and DPO quickly plateau in efficiency gains, GRPO using reinforcement learning with execution feedback continuously optimizes code performance. Experiments on the newly introduced Venus dataset and the APPS benchmark demonstrate that GRPO significantly improves both functional correctness (PASS@1 from 47% to 62%) and efficiency compared to human solutions (BEYOND-I from 31% to 45%). The work addresses a critical gap in LLM-generated code, where models often produce functionally correct but computationally inefficient solutions, and shows that online reinforcement learning with direct execution feedback enables models to achieve sustained improvements in code efficiency through iterative test-time optimization.

**Questions:**

- Could the authors provide a pointer to the reproducible artifact of Venus, Afterburner, and Monolith?

**Ethical Concerns:**

["NO or VERY MINOR ethics concerns only"]

**Final Justification:**

I am glad to see that the authors commit to releasing their data and artifacts, which was my major concern. I am keeping my score to 5 to support this paper's acceptance.

**Limitations:**

Yes.

**Paper Formatting Concerns:**

No formatting concerns.

**Quality:**

3

**Strengths And Weaknesses:**

## Strength

+ This paper is among the earliest works that aim to improve the "efficiency" of LLM-generated code, not only its correctness. Efficiency is an important yet under-explored perspective of LLM-generated code
+ This paper implements a close-loop pipeline that can automatically generate code, reliably collect the efficiency feedback, and iteratively refine its solutions with Afterburner and Monolith
+ This paper releases the first dataset, Venus, that can be used for training LLMs for efficiency optimization, further boosting the future research of improve the efficiency of LLM-generated code.
+ This paper comprehensively and systematically studied the impacts of SFT, DPO, and GRPO for code efficiency refinement training, providing practical insights for the efficiency optimization training recipe.
+ This paper successfully trained Afterburner, which has only 3B parameters while matching much larger models, such as Qwen-2.5-7B-instruct, and Llama-4-Scout, in code efficiency.

## Weaknesses

Overall, I am very positive about this paper, so I do not have much to complain about. One minor thing is that, while the paper replied "Yes" in their checklist about the release of their data and code, I could not find a link to these artifacts. This discrepancy might downgrade the transparency and reproducibility.

---

> ### Author Rebuttal · Authors · 2025-07-27
>
> Dear Reviewer xncp,
>
> Thank you for your positive feedback!
>
> We have prepared the artifacts of Venus, Afterburner, and Monolith. Due to the new rebuttal policy this year, we cannot share external links at this time. We are committed to reproducibility and will update these artifacts in the final version immediately after the review process.
>
> We hope it can address your concern. Thank you!

---

> ### Comment · Reviewer_xncp · 2025-08-05
>
> I am glad to see that the authors commit to release their data and artifacts. I am keeping my score to 5.

---

> > ### Author Response · Authors · 2025-08-05
> >
> > Dear Reviewer xncp,
> >
> > Thank you for your strong support and positive feedback. We will release our code and data after the review period. We sincerely appreciate the time and effort you dedicated to our manuscript!
> >
> > Best regards,
> > Authors

---

### Official Review · Reviewer_ovks · 2025-07-08

**Clarity:** 3
**Significance:** 3
**Originality:** 2
**Rating:** 4
**Confidence:** 5

**Summary:**

This paper proposes a test-time computation method to iteratively improve the efficiency of code snippets generated by LLMs. Specifically, the authors apply reinforcement learning to fine-tune a Qwen-2.5 3B model. Once the model is sufficiently fine-tuned, it is used to optimize code snippets by leveraging execution feedback. To evaluate the effectiveness of the proposed approach, experiments are conducted on the Venus Python dataset. The authors define three metrics: PASS@1, BEYOND-T, BEYOND-M, and BEYOND-I.
The results demonstrate that the method significantly enhances code efficiency. Moreover, the results also demonstrate that LLM-generated code may be more efficient than human-written code.

**Questions:**

1. Does the proposed dataset contain ground truth code snippets? If not, the reliability of the LLM-generated test cases may be questionable.

2. Are all the collected code snippets correct? For example, if some incorrect snippets simply return without executing any statements, they could achieve perfect efficiency but be meaningless.

3. After fine-tuning on the Venus dataset, can the fine-tuned model generalize and perform well on other datasets?

4. Did you comapre the proposed approach with native "SFT" baseline, which just finetune the model to generate most efficient correct code snippets.

5. Can your proposed algorithm generalize to other base model architectures, such as Llama?

6. Where do the test cases used in the iterative improvement stage come from?

7. When computing the efficiency metric during evaluation, did you filter out incorrect code snippets, or did you only consider correct ones?

**Ethical Concerns:**

["NO or VERY MINOR ethics concerns only"]

**Final Justification:**

Based on the rebuttal, I updated my score. However, the way authors evaluated (i.e., using the same set of test cases for improving the model and evaluating them) is still very problematic to me. All my other points are address by the authors during the detailed rebuttal period.

**Limitations:**

1. **Evaluation Metric Rationale**: The rationale behind the chosen evaluation metric requires further discussion. The authors argue that absolute efficiency metrics are avoided due to their sensitivity to hardware configurations and operating systems. However, this justification is unconvincing for several reasons: (1) their method still relies on comparing absolute efficiency metrics; (2) when measured on the same hardware and operating system, efficiency metrics can be fairly compared, and system noise can be mitigated through multiple runs; and (3) the proposed metric heavily depends on the distribution of efficiency in human-written code. I suggest that the authors also present results using absolute efficiency metrics.

2. **Source of Inference-Time Test Cases**: The origin of the inference-time test cases is unclear. The proposed method relies on these runtime test cases to provide correct and efficient feedback for self-improvement. However, it is not specified where these test cases come from: Are they the same as the evaluation test cases, or are they generated by GPT at runtime? If they are generated by GPT at runtime, there is a risk of intelligence leakage. In other words, if GPT is used to generate test cases, why not use GPT to generate correct or efficient code directly, especially since the results show GPT may outperform the proposed method?

3. **Generalizability of the Proposed Method**: The generalizability of the proposed method is unclear. (1) It is not demonstrated whether the method can generalize to other model architectures, such as Llama. (2) Although Table 1 shows the proposed method outperforms baseline methods, all baselines have been fine-tuned specifically on the proposed dataset. It remains unclear whether the fine-tuned model can generalize to other datasets. I recommend that the authors conduct dynamic evaluation [1, 4] or test on other datasets [2, 3] to better demonstrate generalizability.

4. **Missing Baseline**: Why was a simple baseline—fine-tuning the base model with only the most efficient code snippets—not considered?

5. It is unclear whether the efficiency metric is measured only on the correct code.

[1] DyCodeEval: Dynamic Benchmarking of Reasoning Capabilities in Code Large Language Models Under Data Contamination. ICML 2025

[2] Effibench: Benchmarking the efficiency of automatically generated code. Nips 2024.

[3] Evaluating language models for efficient code generation. COLM 2024.

[4] DynaCode: A Dynamic Complexity-Aware Code Benchmark for Evaluating Large Language Models in Code Generation

**Quality:**

2

**Strengths And Weaknesses:**

Strength:

+ The paper is well-written and easy to follow.
+ The problem addressed is both significant and important.
+ The evaluation results are promising.


Weakness:

– The rationale behind the evaluation metric requires further discussion.

– Some details about the proposed dataset are missing.

– The generalization ability of the proposed model is unclear.

---

> ### Author Rebuttal · Authors · 2025-07-31
>
> Dear Reviewer ovks,
>
> Thank you for your constructive and insightful feedback. We are encouraged that you found our work valuable. Below, we address per questions and suggestions raised.
>
> > Q1 & Q2: Solution Correctness in Venus
>
> All Venus code snippets are exclusively collected from accepted (including the official ground truth) solutions on Leetcode, meaning they passed all test cases. We agree that this validation process is a critical detail. Due to space constraints, we included the full pipeline in Appendix C.1. For improved clarity, we will add a brief explanation and a direct reference in Section 5.1 Dataset Recipe:
>
> *“Venus Python subset contains 2,181 algorithmic problems, each accompanied by a validated test case generator and an average of 106.6 validated human solutions... For each LLM-generated test case input, we follow the paradigm of Mercury, where we execute them through all collected solutions from LeetCode, and only keep those cases having consistent outputs over all correct solutions. More details can be found in Appendix C.1 Rigorous Validation.”*
>
> > Q3: Model Generalization
>
> We evaluated our Venus-tuned model on another dataset APPS [1]. Unlike existing code efficiency benchmarks such as Mercury [2], EffiBench [3] and Venus, which consist of LeetCode-style questions, the APPS dataset features distinct problem formats. As shown in the tables below, Afterburner demonstrates a consistent performance improvement pattern on the APPS dataset. Please refer to Appendix Figure 9 for a complete visualization. We also conducted an additional experiment on LiveCodeBench [8] to demonstrate the consistent performance gain on the contamination-free benchmark.
>
> | Afterburner Base on APPS Iterations | 0 | 1 | 3 | 5 | 7 | 10 |
> | :--- | :--- | :--- | :--- | :--- | :--- | :--- |
> | Pass | 10.67 | 11.67 | 13.67 | 15.00 | 15.33 | 15.67 |
> | Beyond-T | 4.14 | 4.14 | 4.14 | 4.23 | 4.53 | 4.53 |
>
> | Afterburner GRPO on APPS Iterations | 0 | 1 | 3 | 5 | 7 | 10 |
> | :--- | :--- | :--- | :--- | :--- | :--- | :--- |
> | Pass | 13.00 | 15.00 | 20.33 | 24.00 | 28.00 | 31.67 |
> | Beyond-T | 4.69 | 9.04 | 12.36 | 14.85 | 16.01 | 16.18 |
>
> | Afterburner Base on LiveCodeBench (release_v6) Iterations | 0 | 1 | 3 | 5 | 7 | 10 |
> | :--- | :--- | :--- | :--- | :--- | :--- | :--- |
> | Pass % | 5.49 | 6.59 | 6.59 | 7.14 | 7.69 | 7.69 |
>
> | Afterburner GRPO on LiveCodeBench (release_v6) Iterations | 0 | 1 | 3 | 5 | 7 | 10 |
> | :--- | :--- | :--- | :--- | :--- | :--- | :--- |
> | Pass % | 5.49 | 7.69 | 9.89 | 11.54 | 12.09 | 13.19 |
>
> We acknowledge that we missed the reference to the APPS results in the main body. Thank you for pointing it out! We will add [5,6] into related work and append the following statement in Section 6.2 to reference these findings in the main text:
>
> *“...To verify whether our models can generalize to out-of-distribution questions, we evaluated Afterburner on the APPS dataset [1]. As illustrated in Appendix Figure 9, Afterburner demonstrates a similar pattern of performance improvement, confirming its effectiveness on problems with distinct data distribution.”*
>
> > Q4 & L4: Native "SFT" Baseline
> We agree that a comparison with a native SFT baseline is valuable. We interpret "native SFT" in two paradigms and address both below:
>
> - **Code Refinement SFT:** trains a model using (problem_description, original_code) to generate an improved_code. This is conceptually very similar to our Afterburner SFT baseline, which augments the input with performance feedback from the original code. We provide its results in Figure 5.
> - **Direct Generation SFT:** trains a model using (problem_description) to directly generate the improved_code. *We assume your question refers to this paradigm.* We initially omitted this baseline because it lacks a performance feedback mechanism, making a direct comparison with our feedback-driven method is unfair. Furthermore, as noted by prior work Mercury [2], this direct approach can be susceptible to catastrophic forgetting, especially on smaller models.
>
> We agree including the native SFT baseline in this work could serve as a straightforward baseline. Prompted by your suggestion, we have conducted this additional experiment to fine-tune the 'Qwen2.5-3B' model using the same hyperparameters as our main experiments. The results are presented in the table below. We observed model collapse on the native SFT baseline.
> | SFT Paradigm      |  Iteration |   0   |   1   |   3   |   5   |   7   |   10  |
> |--|:---:|:-----:|:-----:|:-----:|:-----:|:-----:|:-----:|
> | Code Refinement   |   Pass %   | 46.00 | 46.00 | 47.00 | 48.33 | 48.67 | 48.67 |
> | | Beyond-I % | 21.01 | 21.09 | 21.88 | 22.31 | 22.44 | 22.50 |
> | Direct Generation |   Pass %   | 21.67 | 21.67 | 22.33 | 22.33 | 22.67 | 22.67 |
> | | Beyond-I % |  5.14 |  5.76 |  6.47 |  6.63 |  6.63 |  6.75 |
>
> > Q5 & L3: Model Generalization
>
> Due to the rebuttal time limitation, we conducted additional experiments on another model ‘Qwen2.5 7B’ and 'GPT-4o'.  As the data shows, larger models do achieve better performance through the IOF iterations. However, the improvement in these vanilla models is significantly less than that of our Afterburner models. This demonstrates that while the IOF framework itself is beneficial, our proposed training paradigm are the primary drivers of the substantial performance gains we report. For other model architectures, such as Llama, we will complete its experiment results in the revision.
>
> | **Qwen2.5 7B Iteration** | 0 | 1 | 2 | 5 | 7 | 10 |
> |---|:---:|:---:|:---:|:---:|:---:|:---:|
> | **Pass** | 60.67 | 65.67 | 69.33 | 74.67 | 79.67 | 83.67 |
> | **Beyond-T** | 27.67 | 32.87 | 34.22 | 39.50 | 41.91 | 42.86 |
> | **Beyond-M** | 29.79 | 31.55 | 36.48 | 40.97 | 43.26 | 44.51 |
> | **Beyond-I** | 21.02 | 28.10 | 33.85 | 37.54 | 37.54 | 38.66 |
>
> | **GPT-4o Iterations** | 0 | 1 | 2 | 4 | 8 |
> |---|:---:|:---:|:---:|:---:|:---:|
> | **Pass %** | 82.26 | 84.00 | 84.33 | 84.33 | 85.33 |
> | **Beyond-T %** | 38.22 | 40.16 | 42.02 | 44.16 | 46.17 |
> | **Beyond-M %** | 42.09 | 43.28 | 44.71 | 46.20 | 48.01 |
> | **Beyond-I %** | 28.89 | 30.60 | 32.55 | 35.02 | 41.29 |
>
> > Q6 & L2: Where do the test cases used in the iterative improvement stage come from?
>
> We use the same pre-generated test cases for both iterative improvement and evaluation. This approach is valid because the model only receives high-level execution feedback—such as pass status, running time, and memory usage—without ever seeing the specific contents of the test cases. This design prevents information leakage and ensures a fair evaluation.
>
> > Q7: When computing the efficiency metric during evaluation, did you filter out incorrect code snippets, or did you only consider correct ones?
>
> During the efficiency evaluation, we didn’t filter out incorrect code. The monolith sandbox will return the default maximum metric values (time: 90 s, memory: 1,048,576 kb, integral: 90 * 1,048,576 = 94371840), so their percentile ranks will be 0. You can consider these efficiency metrics (time, memory, integral) as efficiency-weighted functional correctness (pass).
>
> > L1: Evaluation Metric Rationale.
>
> Thank you for this insightful suggestion! We agree that using absolute efficiency metrics has its advantages, especially 1) it is easy to calculate; 2) it doesn't rely on the distribution of human solutions; 3) it may work if we evaluate all models on the same machine.  We appreciate your perspective and we have revised the paper to clarify our rationale per your suggestion:
>
> - **Benchmark Standardization**: We follow the relative efficiency metrics used in most established code efficiency benchmarks, such as Mercury [2], EffiBench [3], EvalPerf [4] and ENAMEL [7].
> - **Result Comparability:** Since Venus is a code efficiency benchmark, we hope the reported score is comparable. While absolute efficiency metrics may be infeasible to compare across machines, relative metrics can handle this issue. For example, a solution that is faster than 80% of human solutions on one machine is likely to be similarly performant relative to the same set of solutions on another machine.
> - **Performance Normalization:** Absolute runtimes can vary by orders of magnitude across different test cases. This large range means that a model's aggregate score can be dominated by a few long-running outliers, masking subtle but significant improvements. Our relative metric normalizes performance against the distribution of human-written code, providing a more stable and fine-grained measure of efficiency improvement.
> - **Training vs. Evaluation:** You correctly note that we use absolute metrics internally. We do this specifically during model training, where the raw execution time and memory provide simple and effective reward signals. Since training occurs in an isolated sandbox on a single hardware setup, comparability is not a concern. Furthermore, for reinforcement learning, an unbounded absolute metric allows the model to explore solutions that surpass the efficiency of any known human code. However, for the final evaluation, the need for a normalized, comparable, and robust benchmark makes the relative metric the superior choice.
>
> **We hope our explanation can address your concern. Please ask follow-ups if you have any other questions. Thank you!**
>
> ### Reference
>
> [1] Measuring coding challenge competence with apps.
>
> [2] Mercury: A code efficiency benchmark for code large language models.
>
> [3] Effibench: Benchmarking the efficiency of automatically generated code.
>
> [4] Evaluating language models for efficient code generation.
>
> [5] DyCodeEval: Dynamic Benchmarking of Reasoning Capabilities in Code Large Language Models Under Data Contamination.
>
> [6] DynaCode: A Dynamic Complexity-Aware Code Benchmark for Evaluating Large Language Models in Code Generation
>
> [7] How efficient is llm-generated code? a rigorous & high-standard benchmark.
>
> [8] LiveCodeBench: Holistic and Contamination Free Evaluation of Large Language Models for Code.

---

> > ### Comment · Reviewer_ovks · 2025-08-06
> >
> > I thank the authors for providing a detailed rebuttal. It clears out many of my concerns. However, I am still worried about the following points:
> >
> > "
> > Q6 & L2: Where do the test cases used in the iterative improvement stage come from?
> >
> > We use the same pre-generated test cases for both iterative improvement and evaluation. This approach is valid because the model only receives high-level execution feedback—such as pass status, running time, and memory usage—without ever seeing the specific contents of the test cases. This design prevents information leakage and ensures a fair evaluation.
> > "
> >
> > I do not think this approach has a fair evaluation, as you are biasing the model towards these test case signals and again evaluating based on them.
> >
> > "During the efficiency evaluation, we didn’t filter out incorrect code. The monolith sandbox will return the default maximum metric values (time: 90 s, memory: 1,048,576 kb, integral: 90 * 1,048,576 = 94371840), so their percentile ranks will be 0. You can consider these efficiency metrics (time, memory, integral) as efficiency-weighted functional correctness (pass)."
> >
> > I think this somewhat undermines the pure efficiency gain. In the later version of the paper, it might be good to also evaluate the efficiency gain of the correct samples.

---

> ### Author Response · Authors · 2025-08-07
> **Further Responses [Part 1/2]**
>
> Dear Reviewer ovks,
>
> Thank you for your insightful follow-up questions. We are pleased that our previous response resolved most of your concerns, and we appreciate the opportunity to provide further clarification.
>
> > Q6 & L2: Where do the test cases used in the iterative improvement stage come from?
>
> You raised an important point regarding the source of test cases used during the iterative inference stage. We'd like to clarify our original process and present a new experiment to demonstrate its robustness.
>
> In the current iterative inference stage, we ran pre-generated 100 test cases on the original solution and fed its **holistic performance metrics** (Passed, Time, Memory, Integral) back into the Afterburner prompt template. For each iteration, we aggregate these metrics as the evaluation results. Notably, **since ‘Passed’ is True if and only if the original solution passed all the test cases**, Afterburner is not able to get individual test case information and hardly overfit all of them during the inference stage. In the real-world applications, we can further eliminate the potential bias (or overfitting) by increasing the number of test cases.
>
> ```
> <…>
> ## Problem Description
> {problem_description}
>
> ## Original Solution
> {original_solution}
>
> ## Original Performance
> Passed: {original_passed} / Time: {original_time} / Memory: {original_memory} / Integral: {original_integral}
> <…>
> ```
>
> While the chance of test case leakage is rare, **we agree that using different test cases for inference and evaluation is more rigorous**. To address this concern, we set the existing test cases in the dataset as the public test cases. Then we follow the setup of EffiLearner [2] to generate 100 private test cases using corresponding test case generators. To avoid potential data leakage, we also filter out identical cases in the public test cases.
> During the iterative optimization stage, we use the public test cases to get the overhead information and optimize the inefficient code. After generating the improved code, we then use the private test cases to measure the performance of the improved code. The evaluation results are shown below:
>
> ### Afterburner-GRPO (inference on public test cases and evaluation on public test cases)
> | **Iteration** | **0** | **1** | **2** | **3** | **4** | **5** | **6** | **7** | **8** | **9** | **10** |
> |---|:---:|:---:|:---:|:---:|:---:|:---:|:---:|:---:|:---:|:---:|:---:|
> | Pass % | 47.33 | 50.33 | 52.00 | 54.50 | 57.00 | 58.17 | 59.34 | 60.50 | 61.18 | 61.67 | 61.67 |
> | Beyond-T % | 31.22 | 36.69 | 38.41 | 39.64 | 40.83 | 42.35 | 43.49 | 44.3 | 44.82 | 45.17 | 45.17 |
> | Beyond-M % | 25.14 | 30.21 | 34.17 | 37.45 | 40.72 | 42.55 | 44.39 | 46.22 | 47.65 | 48.05 | 48.05 |
> | Beyond-I % | 18.24 | 24.81 | 29.44 | 30.85 | 33.56 | 35.48 | 37.09 | 38.01 | 38.62 | 38.95 | 38.95 |
>
> ### Afterburner-GRPO (inference on public test cases and evaluation on private test cases)
> | **Iteration** | **0** | **1** | **2** | **3** | **4** | **5** | **6** | **7** | **8** | **9** | **10** |
> |---|:---:|:---:|:---:|:---:|:---:|:---:|:---:|:---:|:---:|:---:|:---:|
> | Pass % | 46.67 | 51.00 | 52.67 | 54.00 | 56.00 | 58.33 | 59.67 | 61.00 | 62.00 | 62.00 | 62.00 |
> | Beyond-T % | 29.73 | 35.28 | 38.01 | 38.46 | 39.10 | 40.50 | 40.50 | 42.63 | 43.19 | 43.16 | 44.93 |
> | Beyond-M % | 25.43 | 28.59 | 32.15 | 35.21 | 38.66 | 40.73 | 43.21 | 44.92 | 44.92 | 44.94 | 45.89 |
> | Beyond-I % | 17.85 | 23.24 | 28.76 | 29.95 | 33.27 | 35.10 | 36.99 | 37.62 | 38.50 | 38.50 | 38.50 |
>
> We can observe that the performance between the current evaluation (using private test cases) and the previous evaluation (using public test cases) is very similar. Their Pearson correlations (Pass: 0.9939, Beyond-T: 0.9852, Beyond-M: 0.9955, Beyond-I: 0.9986) demonstrate that our presented evaluation is not biased to the public test cases, i.e., only providing the overhead information to LLMs would not leak the test cases and cause the Afterburner-generated code to be biased to these test cases. We will update these evaluation scores using private test cases in the revision. We hope the results and explanation can address your concern.

---

> > ### Author Response · Authors · 2025-08-07
> > **Further Responses [Part 2/2]**
> >
> > Q7: When computing the efficiency metric during evaluation, did you filter out incorrect code snippets, or did you only consider correct ones?
> >
> > This is an excellent suggestion. Our original metric, which includes all generated code, is designed to provide a normalized comparison across different models, as conditioning on correctness makes the sample set for each model different. However, **we agree that evaluating absolute efficiency gains on functionally correct solutions offers a more isolated view for a single model efficiency optimization.**
> >
> > We managed to aggregate the absolute efficiency gain on the subset of functionally correct solutions generated by Afterburner-GRPO. This analysis offers a direct view of pure efficiency improvement.
> >
> > | **Iteration** | **0** | **1** | **2** | **4** | **8** |
> > |---|---:|---:|---:|---:|---:|
> > | Pass | 140 | 153 | 158 | 168 | 186 |
> > | Time (S) | 14308 | 12903 | 11690 | 9354 | 9390 |
> > | Time Average (S) | 102.20 | 84.33 | 73.99 | 55.68 | 50.48 |
> > | Memory (Gb) | 157 | 156 | 141 | 128 | 119 |
> > | Memory Average (Gb) | 1.12 | 1.02 | 0.89 | 0.76 | 0.64 |
> >
> > We sincerely thank you for your constructive feedback, which has helped us strengthen the quality of our work. We will incorporate these new results and discussions into our revision, **including details on the Venus dataset, model generalization performance, and different prespectives of model evaluation**.
> >
> > If you have any other questions, we are happy to provide further clarification needed.

---

### Note · Authors · 2025-08-13

We wish to extend our sincere gratitude for your thoughtful and constructive feedback. The time and expertise you dedicated to our work have been instrumental in significantly improving our manuscript. **We are pleased to report that our rebuttal responses were positively received by all reviewers.**

## Rebuttal Summary:

- **Reviewer ovks:** The reviewer confirmed that “it clears out many of my concerns”.
- **Reviewer xncp:** The reviewer was satisfied with our commitment to a public release upon acceptance and maintained their positive rating.
- **Reviewer z3gd:** We clarified our methodology regarding bootstrap sampling, hyperparameter selection, and validation on the contamination-free benchmark. These clarifications resolved the reviewer's primary concerns and they accordingly increased their rating.
- **Reviewer UfKJ:** In response to the reviewer's feedback, we conducted extensive new experiments to demonstrate our framework's generalization to larger models (Qwen-2.5-7B) and to verify that RL-based training is the key performance driver. Moreover, we also explored the history-aware framework. The reviewer was satisfied with the thorough rebuttal and raised their score.

## Major Discussion:

- **Model Generalizability** (Reviewers ovks and UfKJ): To substantiate the broader applicability of our work, we have now validated Afterburner on larger models (Qwen-2.5-7B) and across distinct benchmarks (APPS, LiveCodeBench).
- **Evaluation Rationale** (Reviewer ovks): We provided results on held-out test cases and reported absolute efficiency gains to further strengthen our evaluation protocol.
- **Experimental Details** (Reviewer z3gd): We incorporated detailed clarifications on hyper-parameter selection and statistical significance.
- **Baseline Comparisons** (Reviewer UfKJ): We conducted baseline evaluation on *Qwen-2.5-3B, Qwen-2.5-7B, and OpenAI GPT-4o.*
- **Future Directions** (Reviewer UfKJ): Inspired by the reviewer, we explored a history-aware framework and will incorporate the efficiency spectrum into the revision.

We believe these revisions have substantially strengthened the paper and comprehensively addressed all major concerns. Thank you again for the invaluable feedback.

---

### Decision · Program_Chairs · 2025-09-17

**Decision:**

Accept (poster)

**Comment:**

The paper proposes a test-time iterative optimization framework for code generation, where an LLM refines code in a closed-loop system using empirical performance feedback from an execution sandbox. Three training strategies are explored: SFT, DPO, and GRPO. The GRPO variant with execution feedback yields the strongest gains, improving both pass@1 and the likelihood of outperforming human submissions in efficiency.

Strengths:
- Addresses an important and impactful problem (ovks, xncp, z3gd)
- Implements a closed-loop pipeline that integrates code generation, efficiency feedback collection, reward formulation, and iterative refinement (xncp, z3gd)
- Promising evaluation results with comprehensive experiments across training strategies (xncp, ovks, z3gd, UfKJ)

Weaknesses:
- The rationale behind the evaluation metric needs further discussion, though the authors clarified during rebuttal (ovks)
- Some dataset details were missing and the authors reasonably addressed in rebuttal (ovks)
- Open questions remain on scaling the method to larger models (UfKJ)

This work presents a well-motivated and carefully evaluated approach for integrating performance/efficiency feedback into code generation via iterative optimization. While some methodological details and scaling considerations remain open, the contribution is solid and the findings are empirically supported.